# Measuring changes in *Plasmodium falciparum* census population size in response to sequential malaria control interventions

Kathryn E Tiedje[1,2†], Qi Zhan[3,4†], Shazia Ruybal-Pésantez[2‡], Gerry Tonkin-Hill[2,5], Qixin He[4§], Mun Hua Tan[1], Dionne C Argyropoulos[1#], Samantha Deed[1,2], Anita Ghansah[6], Oscar Bangre[7], Abraham R Oduro[7], Kwadwo A Koram[8], Mercedes Pascual[9,10], Karen P Day[2*]

[1]Department of Microbiology and Immunology, Bio21 Institute and Peter Doherty Institute for Infection and Immunity, The University of Melbourne, Melbourne, Australia; [2]School of BioSciences, Bio21 Institute, The University of Melbourne, Melbourne, Australia; [3]Committee on Genetics, Genomics and Systems Biology, The University of Chicago, Chicago, United States; [4]Department of Ecology and Evolution, The University of Chicago, Chicago, United States; [5]Bioinformatics Division, Walter and Eliza Hall Institute, Melbourne, Australia; [6]Department of Parasitology, Noguchi Memorial Institute for Medical Research, University of Ghana, Legon, Ghana; [7]Navrongo Health Research Centre, Ghana Health Service, Navrongo, Ghana; [8]Epidemiology Department, Noguchi Memorial Institute for Medical Research, University of Ghana, Legon, Ghana; [9]Department of Biology and Department of Environmental Sciences, New York University, New York, United States; [10]Santa Fe Institute, Santa Fe, United States

**\*For correspondence:**
karen.day@unimelb.edu.au

[†]These authors contributed equally to this work

**Present address:** [‡]Department of Infectious Disease Epidemiology and MRC Centre for Global Infectious Disease Analysis, School of Public Health, Imperial College, London, United Kingdom; [§]Department of Biological Sciences, Purdue University, West Lafayette, United States; [#]Infection and Global Health Division, Walter and Eliza Hall Institute of Medical Research, Parkville, Australia; Department of Medical Biology, The University of Melbourne, Melbourne, Australia

**Competing interest:** The authors declare that no competing interests exist.

## eLife Assessment

This **valuable** study highlights how the diversity of the malaria parasite population diminishes following the initiation of effective control interventions but quickly rebounds as control wanes. It also demonstrates that the asymptomatic reservoir is unevenly distributed across host age groups. The data presented are **convincing** and the work shows how genetic studies could be used to monitor changes in disease transmission.

**Abstract** Here, we introduce a new endpoint 'census population size' to evaluate the epidemiology and control of *Plasmodium falciparum* infections, where the parasite, rather than the infected human host, is the unit of measurement. To calculate census population size, we rely on a definition of parasite variation known as multiplicity of infection ($MOI_{var}$), based on the hyper-diversity of the *var* multigene family. We present a Bayesian approach to estimate $MOI_{var}$ from sequencing and counting the number of unique DBLα tags (or DBLα types) of *var* genes, and derive from it census population size by summation of $MOI_{var}$ in the human population. We track changes in this parasite population size and structure through sequential malaria interventions by indoor residual spraying (IRS) and seasonal malaria chemoprevention (SMC) from 2012 to 2017 in an area of high, seasonal malaria transmission in northern Ghana. Following IRS, which reduced transmission intensity by >90% and decreased parasite prevalence by ~40–50%, significant reductions in *var* diversity,

$MOI_{var}$, and population size were observed in ~2000 humans across all ages. These changes, consistent with the loss of diverse parasite genomes, were short-lived and 32 months after IRS was discontinued and SMC was introduced, *var* diversity and population size rebounded in all age groups except for the younger children (1–5 years) targeted by SMC. Despite major perturbations from IRS and SMC interventions, the parasite population remained very large and retained the *var* population genetic characteristics of a high-transmission system (high *var* diversity; low *var* repertoire similarity), demonstrating the resilience of *P. falciparum* to short-term interventions in high-burden countries of sub-Saharan Africa.

## Introduction

Malaria in high-transmission endemic areas of sub-Saharan Africa (SSA) is characterised by vast diversity of the *Plasmodium falciparum* parasites from the perspective of antigenic variation (*Chen et al., 2011*; *Day et al., 2017*; *Otto et al., 2019*; *Ruybal-Pesántez et al., 2022*; *Ruybal-Pesántez et al., 2017*). As with other hosts of hyper-variable pathogens (*Futse et al., 2008*), children experiencing clinical episodes of malaria eventually become immune to disease but not to infection. This results in a large reservoir of chronic asymptomatic infections, in hosts of all ages, sustaining transmission to mosquitos. Given the goal of malaria eradication by 2050, it is therefore of interest to examine how the parasite population changes following perturbation by major intervention efforts, both in terms of its size and underlying population genetics.

So, what do we mean by the parasite population size in the case of *P. falciparum* and how do we measure it? Parasite prevalence, detected by microscopy or more sensitive molecular diagnostics (e.g. PCR), describes the proportion of infected human hosts. Studies of *P. falciparum* genetic diversity have shown that the majority of people in high-transmission endemic areas harbour diverse multiclonal infections measured as the complexity or multiplicity of infection (MOI) (e.g. *Anderson et al., 2000*; *Paul et al., 1995*; *Smith et al., 1999*; *Sumner et al., 2021*) with complex population dynamics (*Bruce et al., 2000*; *Farnert et al., 1997*). These genetic data indicate much larger parasite population sizes than observed by prevalence of infection alone. Thus, from an ecological perspective, we can consider a human host as a patch carrying a number of 'antigenically distinct infections' of *P. falciparum*. The sum of these antigenically distinct infections over all sampled hosts provides us with a census of the parasite count of relevance to monitoring and evaluating malaria interventions. We refer to this census population size hereafter simply as population size but make clear that this measure is distinct from effective population size ($N_e$) as measured by neutral variation. This count can be scaled from the host sample to the larger denominator of a host population in the area of interest.

Diversity of *P. falciparum* single copy surface antigen genes such as circumsporozoite protein (*csp*), merozoite surface protein 1 (*msp1*) or 2 (*msp2*), and apical membrane antigen 1 (*ama1*) have each been widely used to measure MOI (e.g. *Falk et al., 2006*; *Lerch et al., 2017*; *Nelson et al., 2019*). They have become part of newer genetic panels (e.g. Paragon v1 [*Tessema et al., 2022*] and AMPLseq v1 [*LaVerriere et al., 2022*]) specifically for MOI determination. Typically, MOI is reported as the maximum number of alleles or single locus haplotypes present at the most diverse of these antigen-encoding loci rather than the number of unique multilocus haplotypes of these genes combined, as it is challenging to accurately reconstruct or phase these haplotypes in hosts with an MOI>3 (*Lerch et al., 2019*). Each of these genes is under balancing selection with a few geographically common haplotypes and many very rare haplotypes in moderate- to high-transmission settings (*Markwalter et al., 2022*; *Sumner et al., 2021*). Where there is a high probability of co-occurrence of two or more common single locus haplotypes in a host, genotyping each of these single copy antigen genes alone will underestimate MOI. Single nucleotide polymorphism (SNP) panels have been used to define the presence of multiclonal infections with limited reliability to estimate MOI for highly complex infections, typical in high transmission, even with the use of computational methods (*Labbé et al., 2023*).

As an alternative to genotyping single copy antigen genes and biallelic SNP panels to estimate MOI, we have proposed the use of a fingerprinting methodology known as *var*coding to genotype the hyper-diverse *var* multigene family (~50–60 *var* genes per haploid genome) (*Day et al., 2025*). This method employs an ~450 bp region of a *var* gene, known as a DBLα tag encoding the immunogenic Duffy-binding-like alpha (DBLα) domain of *P. falciparum* erythrocyte membrane protein 1 (PfEMP1), the major surface antigen of the blood stages (*Zhang and Deitsch, 2022*). Bioinformatic analyses of

a large database of exon 1 sequences of *var* genes showed a predominantly 1-to-1 DBLα-*var* relationship, such that each DBLα tag typically represents a unique *var* gene, especially in high transmission (*Tan et al., 2023*). The extensive diversity of DBLα tags, together with the very low percentage of *var* genes shared between parasites (*Chen et al., 2011*; *Day et al., 2017*; *Ruybal-Pesántez et al., 2022*; *Ruybal-Pesántez et al., 2017*), facilitates measuring MOI by amplifying, pooling, sequencing, and counting the number of unique DBLα tags (or DBLα types) in a host (*Ruybal-Pesántez et al., 2022*; *Tiedje et al., 2022*). From a single PCR with degenerate primers and amplicon sequencing, the method specifically counts the most diverse DBLα types, designated non-upsA, per infection to arrive at a metric we call $MOI_{var}$. It is not based on assigning haplotypes but exploits the fact that *var* repertoires are non-overlapping, especially in high transmission. Instead, it assumes a set number of non-upsA types per genome based on repeated sampling of 3D7 control isolates accounting for PCR sampling errors to calculate $MOI_{var}$ (*Ghansah et al., 2023*; *Ruybal-Pesántez et al., 2022*; *Tiedje et al., 2022*). Consequently, rather than looking at the diversity of a single copy antigen-encoding gene like *csp*, *msp2*, or *ama1* to calculate MOI, by *var*coding we are looking at sets of up to 45 non-upsA DBLα types per genome. Prior work has shown that *var*coding is more sensitive to measure MOI in high transmission where there is an extremely high prevalence of multiclonal infections that cannot be accurately phased with either biallelic SNP panels (*Ghansah et al., 2023*; *Labbé et al., 2023*; *Tessema et al., 2022*; *Watson et al., 2021*) or combinations of single copy antigen genes (*Sumner et al., 2021*).

Here, we report an investigation of changes in parasite census population size and structure through two sequential malaria control interventions between 2012 and 2017 in Bongo District located in the Upper East Region of northern Ghana, one of the 12 highest burden countries in Africa (*World Health Organization, 2022*). We present a novel Bayesian modification to the published *var*coding approach (*Ghansah et al., 2023*; *Ruybal-Pesántez et al., 2022*; *Tiedje et al., 2022*) that takes into account under-sampling of non-upsA DBLα types in an isolate to estimate $MOI_{var}$ (*Ruybal-Pesántez et al., 2022*; *Tiedje et al., 2022*) and therefore population size. We document *P. falciparum* prevalence, as well as *var* diversity and population structure from baseline in 2012 through a major perturbation by a short-term indoor residual spraying (IRS) campaign managed under operational conditions, which reduced transmission intensity by >90% as measured by the entomological inoculation rate (EIR) and decreased parasite prevalence by ~40–50% (*Tiedje et al., 2022*). Next, we followed what happened to parasite population size more than two years after the IRS intervention was discontinued and seasonal malaria chemoprevention (SMC) was introduced for children between the ages of 3 and 59 months (i.e. <5 years) (*Wagman et al., 2018*). Detectable changes in parasite population size were seen as a consequence of the IRS intervention, but this quantity rapidly rebounded 32 months after the intervention ceased. Overall, throughout the IRS, SMC, and subsequent rebound, the parasite population in humans remained large in size and retained the *var* population genetic characteristics of high transmission (i.e. high *var* diversity, low *var* repertoire overlap), demonstrating the overall resilience of the species to survive significant short-term perturbations.

## Results

Between 2013 and 2015, three rounds of IRS with non-pyrethroid insecticides were implemented across all of Bongo District (*Figure 1A*). Coincident with the >90% decrease in transmission following IRS (*Tiedje et al., 2022*), the prevalence of microscopic *P. falciparum* infections compared to the 2012 baseline survey (pre-IRS) declined by 45.2% and 35.7% following the second (2012–2014) and the third (2012–2015) round of IRS, respectively (*Figure 1B*, *Appendix 1—table 1*). These declines in parasite prevalence were observed across all ages, with the greatest impacts being observed on the younger children (1–5 years) who were ~3 times less likely to have an infection in 2015 (post-IRS) compared to 2012 (pre-IRS) (*Figure 1C*, *Appendix 1—table 1*). These reductions were, however, short-lived and in 2017, 32 months after the discontinuation of IRS, but during SMC, overall *P. falciparum* prevalence rebounded to 41.2%, or near pre-IRS levels (*Figure 1B*, *Appendix 1—table 1*). Importantly, this increase in the prevalence of infection in 2017 was only observed among the older age groups (i.e. ≥6 years) (*Figure 1C*, *Appendix 1—table 1*). This difference by age group in 2017 can be attributed to SMC, which only targets children between 3 and 59 months (i.e. <5 years). A notable increase in parasite prevalence for adolescents (11–20 years) and adults (>20 years) was found in 2017 relative to 2012 (pre-IRS) (*Figure 1C*, *Appendix 1—table 1*).

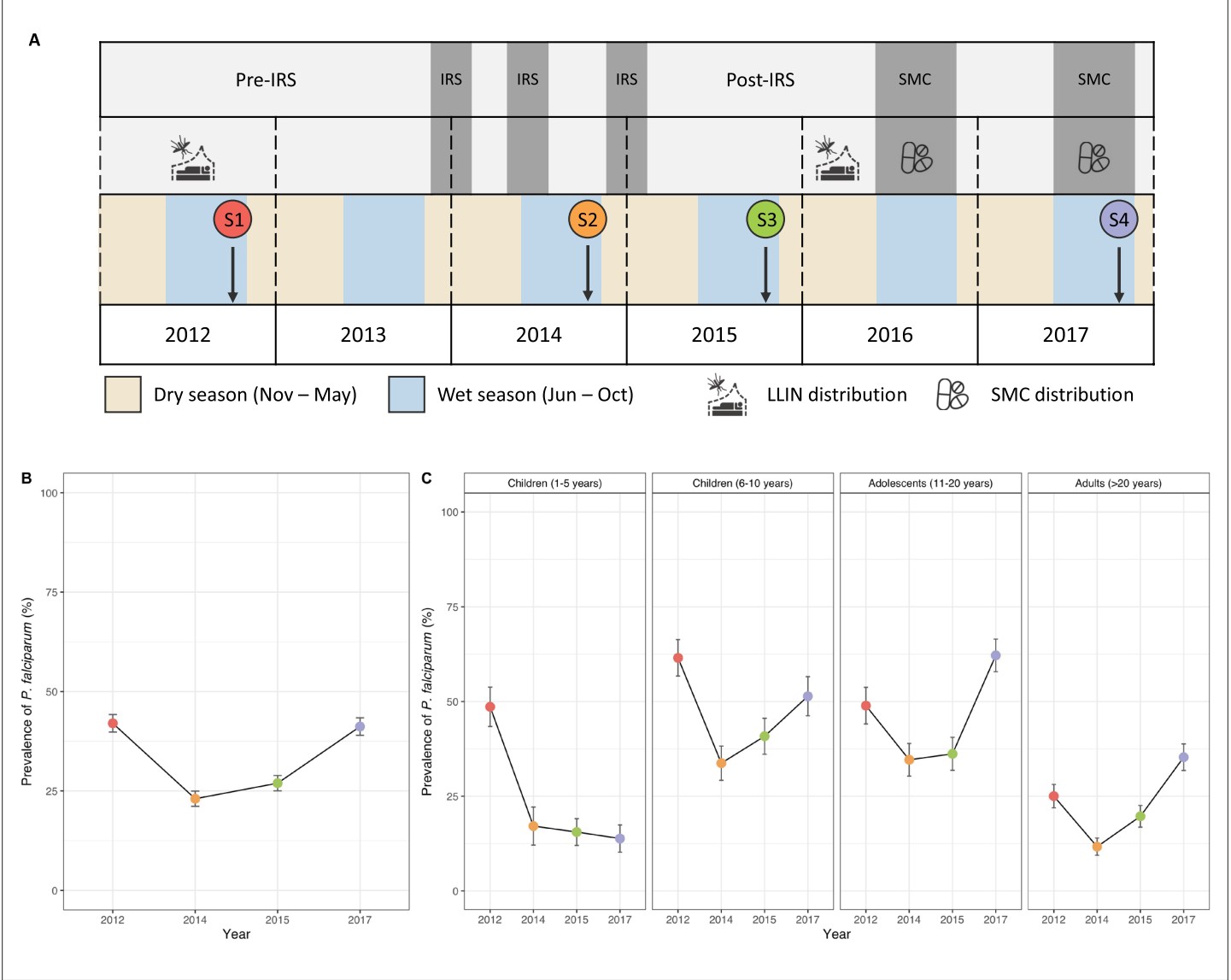

**Figure 1.** Study design and changes in the prevalence of microscopic *P. falciparum* infection following the indoor residual spraying (IRS) and seasonal malaria chemoprevention (SMC) interventions in Bongo, Ghana. (**A**) Four age-stratified cross-sectional surveys of ~2000 participants per survey were conducted in Bongo, Ghana, at the end of the wet seasons in October 2012 (Survey 1, baseline pre-IRS, red), October 2014 (Survey 2, during IRS, orange), October 2015 (Survey 3, post-IRS, green), and October 2017 (Survey 4, SMC, purple) (see Materials and methods, *Appendix 1—table 1*). The three rounds of IRS (grey areas) were implemented between 2013 and 2015 (*Tiedje et al., 2022*). SMC was distributed to all children <5 years of age during the wet seasons in 2016 (two rounds between August and September 2016) and 2017 (four rounds between September and December 2017) (*Gogue et al., 2020*). Both IRS and SMC were implemented against a background of widespread long-lasting insecticidal net (LLIN) usage (*Tiedje et al., 2022*). This figure was adapted from *Tiedje et al., 2022*, Figure 1 (CC BY 4.0 licence). The copyright holder has granted permission to publish under a CC BY 4.0 licence. Prevalence of microscopic *P. falciparum* infections (%) in the (**B**) study population and (**C**) for all age groups (years) in each survey (*Appendix 1—table 1*). Error bars represent the upper and lower limits of the 95% confidence interval (CI) calculated using the Wald interval.

Next, we wanted to explore changes in population size measured by MOI$_{var}$. As this metric is based on non-overlap of *var* repertoire diversity of individual isolates, specifically non-upsA DBLα types, we investigated whether DBLα isolate repertoire similarity (or overlap), as measured by pairwise type sharing (PTS), increased following the sequential interventions (i.e. IRS and SMC). *Figure 2* shows that median PTS values for both upsA and non-upsA DBLα types remained low in all surveys, although the PTS distributions for both groups changed significantly at each of the study time points relative to the 2012 baseline survey (pre-IRS) (p-values<0.001, Kruskal-Wallis test) (*Figure 2*). Somewhat unexpectedly, the change was in the direction of reduced similarity (i.e. less overlap) with lower median PTS

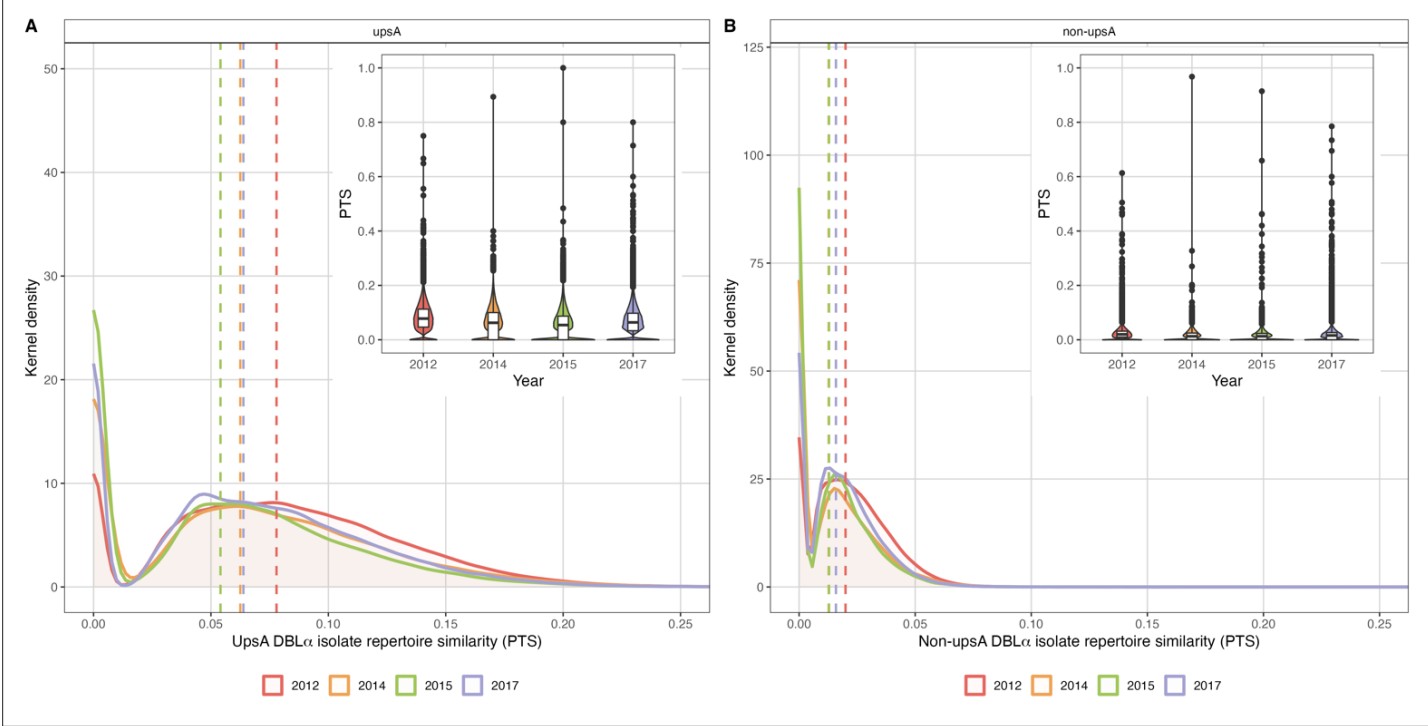

**Figure 2.** Sharing of upsA and non-upsA DBLα types among the DBLα isolate repertoires in 2012 (pre-indoor residual spraying [IRS], red), 2014 (during IRS, orange), 2015 (post-IRS, green), and 2017 (seasonal malaria chemoprevention [SMC], purple). The overlapping density and violin plots (upper right-hand corners) show the distribution of pairwise type sharing (PTS) scores (i.e. DBLα isolate repertoire similarity) between the (**A**) upsA and (**B**) non-upsA DBLα isolate repertoires for those isolates with DBLα sequencing data (*Appendix 1—tables 2 and 3*) in each survey. The PTS scales in the density plots have been zoomed in to provide better visualisation of the upsA and non-upsA DBLα type PTS distributions. The coloured dashed lines in the density plots indicate the median PTS scores in each survey for the upsA (2012 [red]=0.078, 2014 [orange]=0.063, 2015 [green]=0.054, and 2017 [purple]=0.064) and non-upsA (2012 [red]=0.020, 2014 [orange]=0.013, 2015 [green]=0.013, and 2017 [purple]=0.016) DBLα types. *Note:* The non-upsA median PTS values in 2014 (orange) and 2015 (green) were both 0.013 and overlap in the figure. In the PTS violin plots, the central box plots indicate the medians (centre line), interquartile range (IQR, upper and lower quartiles), whiskers (1.5x IQR), and outliers (points).

The online version of this article includes the following figure supplement(s) for figure 2:

**Figure supplement 1.** Sharing of upsA and non-upsA DBLα types among the DBLα isolate repertoires for those isolates with an estimated $MOI_{var}$ equal to one ($MOI_{var}$=1) in 2012 (pre-indoor residual spraying [IRS], red), 2014 (during IRS, orange), 2015 (post-IRS, green), and 2017 (seasonal malaria chemoprevention [SMC], purple).

scores and a larger number of isolates sharing no DBLα types (i.e. PTS = 0) in 2014, 2015, and 2017 compared to 2012. Relevant to the measurement of $MOI_{var}$, the median PTS scores for non-upsA DBLα types were lower following the IRS intervention ($PTS_{non-upsA}$: 2014=0.013 and 2015=0.013 vs. $PTS_{non-upsA}$: 2012=0.020). In 2017, the non-upsA PTS distributions shifted back toward higher median PTS scores ($PTS_{non-upsA}$=0.016) and fewer isolates shared no DBLα types relative to 2014 and 2015 (*Figure 2*). To verify this pattern was not influenced by multiclonal infections ($MOI_{var}$>1), we also examined isolates with monoclonal infections ($MOI_{var}$=1) and found that this non-overlapping structure persisted regardless of infection complexity, particularly for the non-upsA DBLα types (*Figure 2—figure supplement 1*). These PTS data make clear that we were dealing with a large, highly diverse parasite population where 99.9% of the isolate comparisons in all surveys had $PTS_{non-upsA}$ scores ≤0.1 (i.e. shared ≤10% of their non-upsA DBLα types), indicating that DBLα isolate repertoires were highly unrelated (*Figure 2*). In fact, throughout the IRS, SMC, and subsequent rebound, very few DBLα isolate repertoires were observed to be related, with <0.003% isolate comparisons in each survey having a $PTS_{non-upsA}$≥0.5 (i.e. siblings or recent recombinants) (*Figure 2*).

The raw data of non-upsA DBLα isolate repertoire sizes were used to estimate $MOI_{var}$ as adjusted using the Bayesian approach based on pooling the maximum *a posteriori* MOI estimates (*Figure 3*, *Figure 3—figure supplement 1*) and the mixture distribution (*Figure 3—figure supplement 2*). We observed that at baseline in 2012, the majority (89.2%) of the population across all ages carried

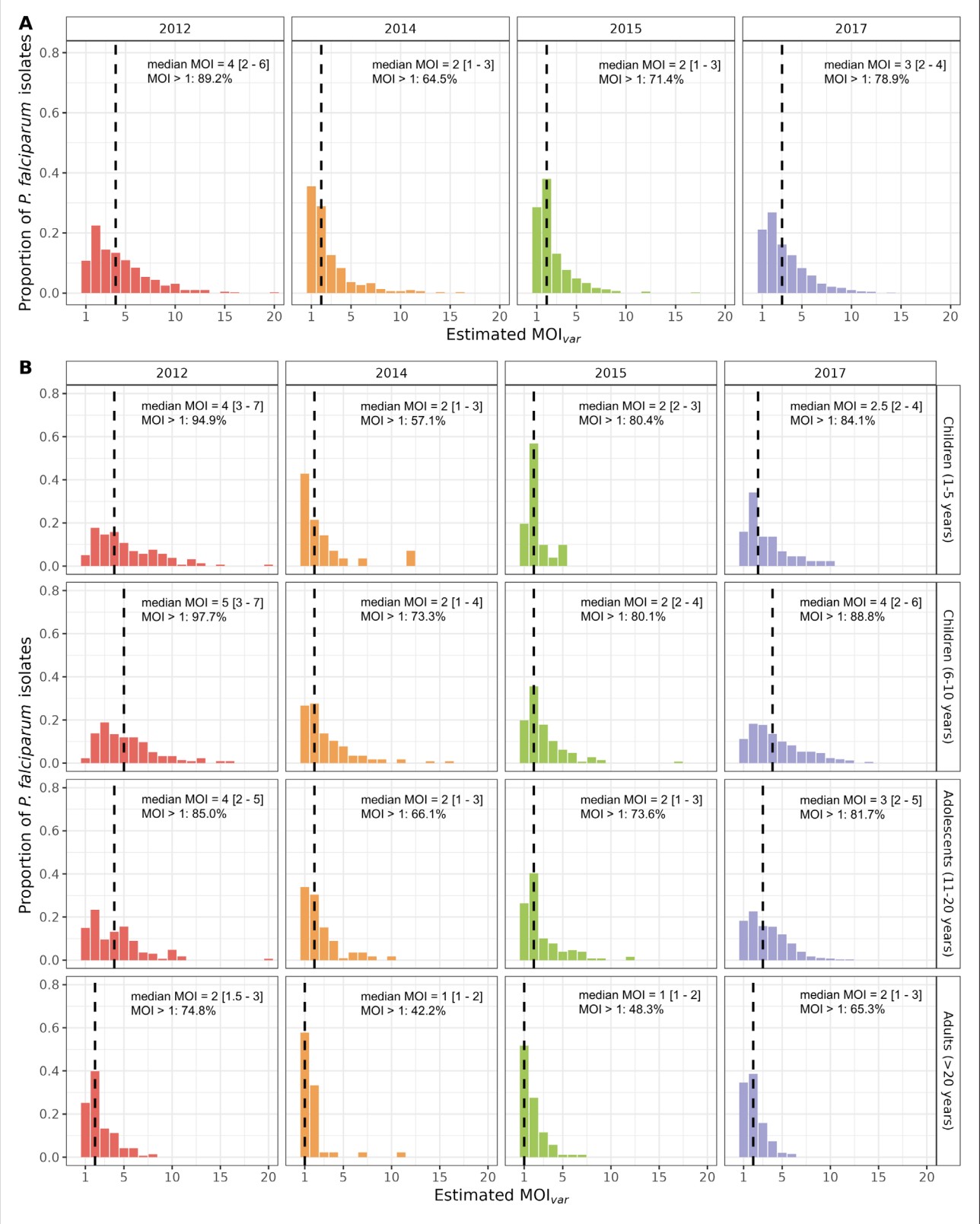

**Figure 3.** MOI$_{var}$ distributions in 2012 (pre-indoor residual spraying [IRS], red), 2014 (during IRS, orange), 2015 (post-IRS, green), and 2017 (seasonal malaria chemoprevention [SMC], purple) based on pooling the maximum *a posteriori* multiplicity of infection (MOI) estimates. Estimated MOI$_{var}$ distributions for the (**A**) study population and (**B**) for all age groups (years) in each survey for those isolates with DBLα sequencing data (*Appendix 1—tables 2 and 3*). The median MOI$_{var}$ values are indicated with the black dashed lines and have been provided in the top right corner (median MOI$_{var}$

*Figure 3 continued on next page*

Figure 3 continued

value [interquartile range (IQR), upper and lower quartiles]) along with the percentage of *P. falciparum* infections that were multiclonal (MOI$_{var}$>1) in each survey and age group (years).

The online version of this article includes the following figure supplement(s) for figure 3:

**Figure supplement 1.** UpsA and non-upsA DBLα isolate repertoire sizes in 2012 (pre-indoor residual spraying [IRS], red), 2014 (during IRS, orange), 2015 (post-IRS, green), and 2017 (seasonal malaria chemoprevention [SMC], purple).

**Figure supplement 2.** MOI$_{var}$ distributions in 2012 (pre-indoor residual spraying [IRS], red), 2014 (during IRS, orange), 2015 (post-IRS, green), and 2017 (seasonal malaria chemoprevention [SMC], purple) based on the mixture distribution approach.

multiclonal infections (median MOI$_{var}$=4 [interquartile range (IQR): 2–6]) (*Figure 3A*). Following the IRS intervention, the estimated MOI$_{var}$ distributions became more positively skewed, indicating that a lower proportion of participants harboured multiclonal infections with a lower median MOI$_{var}$ in 2014 (64.5%; median MOI$_{var}$=2 [IQR: 1–3]) and 2015 (71.4%; median MOI$_{var}$=2 [IQR: 1–3]) compared to 2012 (*Figure 3A*). These reductions in median MOI$_{var}$ and the proportion of multiclonal infections, which were observed across all age groups (*Figure 3B*), are consistent with the >90% decrease in transmission intensity following the IRS in turn reducing exposure to new parasite genomes. However, in 2017, both the median MOI$_{var}$ (3 [IQR: 2–4]) and the proportion of multiclonal infections (78.9%) rebounded in all age groups, even among the younger children (1–5 years) predominantly targeted by SMC (*Figure 3*). While the prevalence of infection in 2017 remained low for the younger children (1–5 years), those infected still carried multiclonal infections (84.1% of those infected) (*Figure 3B*). Although the MOI$_{var}$ distributions across all age groups started to rebound in 2017 (i.e. less positively skewed compared to 2014 and 2015), they had not fully recovered to the 2012 baseline patterns (*Figure 3*). This was most apparent among the younger children (1–5 years), as a larger proportion of isolates in 2017, compared to 2012, had MOI$_{var}$ values equal to one or two, while a smaller proportion had MOI$_{var}$ values ≥5 (*Figure 3B*).

Census population size, measured as the number of *P. falciparum var* repertoires circulating in the population during each survey, was estimated by summation of isolate MOI$_{var}$ (see Materials and methods; *Figure 4*, *Appendix 1—table 2*). In 2014 during IRS, this number decreased by 71.4% relative to the 2012 baseline survey (pre-IRS) (*Figure 4C and E*), whereas prevalence decreased by 54.5% (*Figure 4C and E*). Although census population size increased slightly in 2015 relative to 2014 (*Figure 4A*), there were still 64.4% fewer *var* repertoires in the population compared to 2012 (*Figure 4C and E*) in comparison to a 42.6% decrease in prevalence (*Figure 4C and E*). Importantly, this loss of *var* repertoires in 2014 and 2015 following the IRS intervention was seen for all age groups (*Figure 4B*), with the greatest overall reductions (≥83.8%) being observed for the younger children (1–5 years) (*Figure 4D and F*). However, in 2017, the number of diverse *var* repertoires in the population rebounded, more than doubling between 2015 and 2017 (*Figure 4C and E*). This increase in the number of *var* repertoires was seen for all age groups in 2017, except for the younger children (1–5 years) where those up to 59 months were targeted by SMC (*Figure 4B and D*). In fact, the greatest overall increase was observed for the adolescents (11–20 years) and adults (>20 years), where the number of *var* repertoires in 2017 was ~1.2 times higher compared to 2012 (*Figure 4F*). Similar trends in the number of *var* repertoires were also observed for the older children (6–10 years) in 2017, although the rebound was not as striking as that detected for the adolescents and adults.

As census population size changed considerably during the sequential IRS and SMC interventions, we investigated how the removal or loss of *P. falciparum var* repertoires and subsequent rebound in 2017 altered DBLα type richness, measured as the number of unique upsA and non-upsA DBLα types in the parasite population in each survey. Richness at baseline in 2012 (pre-IRS) was high with a large number of unique DBLα types (upsA = 2218; non-upsA=33,159) (*Figure 5*, *Appendix 1—table 3*) and limited overlap of *var* repertoires (i.e. median PTS$_{non-upsA}$≤0.020) seen in a relatively small study population of 685 microscopically positive individuals (*Figure 2*). In 2014, as *P. falciparum* prevalence and census population size declined (*Figure 4*), so too did the number of DBLα types, resulting in a 32.2% and 55.3% reduction in richness for the upsA and non-upsA DBLα types, respectively, compared to 2012 (*Figure 5*, *Appendix 1—table 3*). Again in 2015, as *P. falciparum* prevalence and population size remained low (*Figure 4*), DBLα type richness was still less than that observed in 2012 (24.6% and 46.0% reduction for upsA and non-upsA DBLα types, respectively) (*Figure 5*, *Appendix 1—table 3*). Finally, in 2017, we found that upsA and non-upsA DBLα type richness rebounded relative to 2014 and

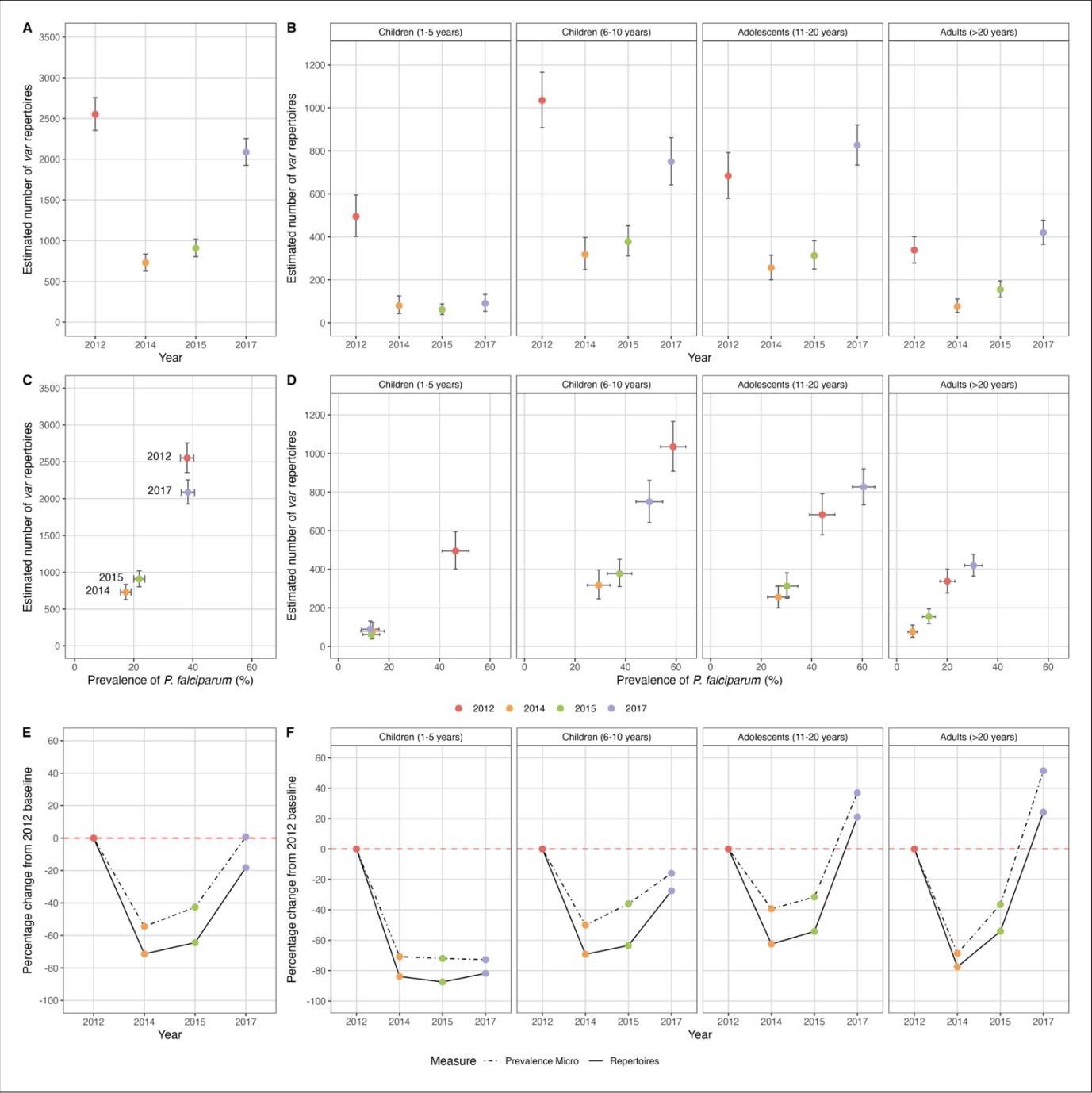

**Figure 4.** Estimated number and relative change in the number of *P. falciparum var* repertoires in 2012 (pre-indoor residual spraying [IRS], red), 2014 (during IRS, orange), 2015 (post-IRS, green), and 2017 (seasonal malaria chemoprevention [SMC], purple). The estimated number of *var* repertoires (i.e. census population size) for those isolates with DBLα sequencing data (*Appendix 1—tables 2 and 3*) in the (**A**) study population and (**B**) for all age groups (years). The estimated number of *var* repertoires vs. *P. falciparum* prevalence for (**C**) study population and (**D**) for all age groups (years) (*Appendix 1—table 2*). The percentage change in *P. falciparum* prevalence (black dotted line) and the estimated number of *var* repertoires (black solid line) in 2014, 2015, and 2017 compared to the 2012 baseline survey (red dashed horizontal line at 0% change) for the (**E**) study population and (**F**) for all age groups (years). Error bars in (**A–D**) represent the upper and lower limits of the 95% confidence intervals (95% CIs). To account for differences in sampling depth across age groups and surveys, we performed subsampling with replacement by selecting the minimum number of individuals in each age group across all surveys. We then calculated the total number of *var* repertoires from these subsampled individuals within each age group in each survey. This approach ensures consistent sample sizes within each age group across all surveys. Finally, we summed the *var* repertoires across age groups to obtain the total *var* repertoire count for each survey. The mean (coloured solid points) and 95% CIs for the number of *var* repertoires were estimated by repeating the subsampling procedure 10,000 times. The CIs were then derived from the distribution of these repeated subsampling replicates. The 95% CIs for *P. falciparum* prevalence (%) were calculated using the Wald interval.

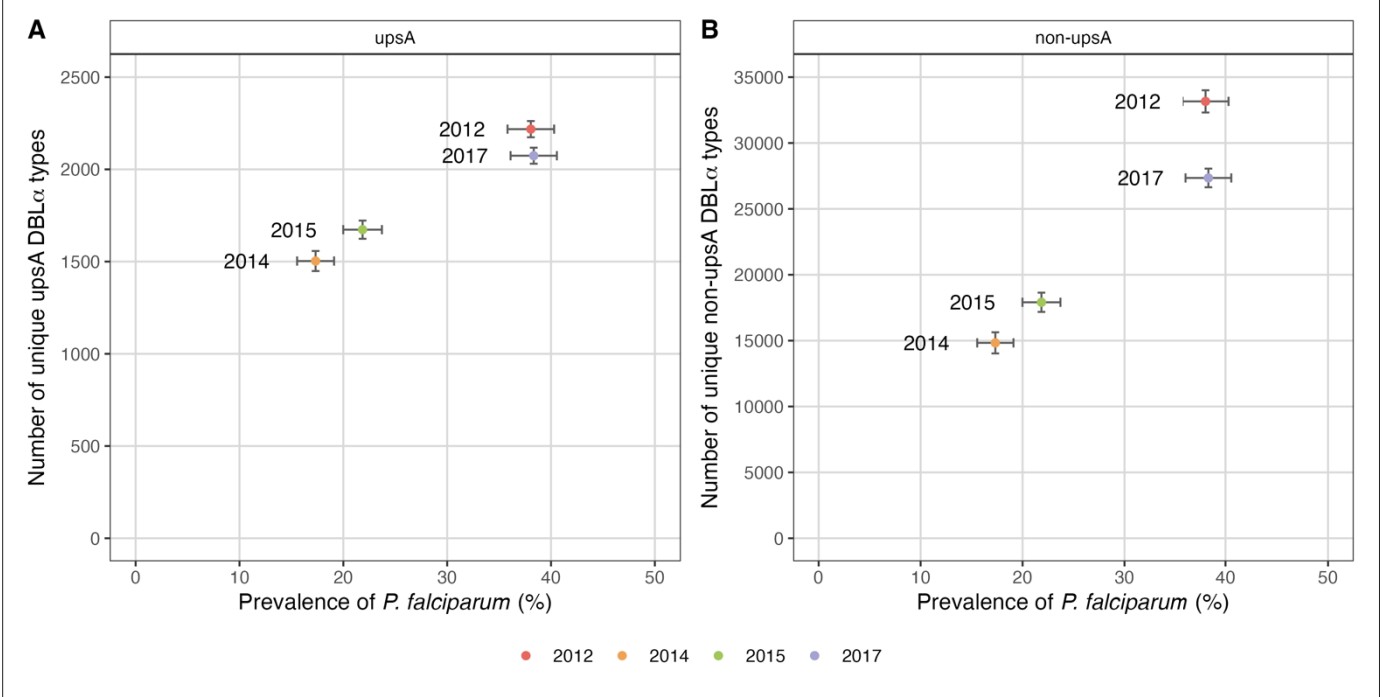

**Figure 5.** UpsA and non-upsA DBLα type richness in 2012 (pre-indoor residual spraying [IRS], red), 2014 (during IRS, orange), 2015 (post-IRS, green), and 2017 (seasonal malaria chemoprevention [SMC], purple). Number of unique (**A**) upsA and (**B**) non-upsA DBLα types (i.e. richness) observed in each survey vs. *P. falciparum* prevalence based on those isolates with DBLα sequencing data (***Appendix 1—tables 2 and 3***). Error bars represent the upper and lower limits of the 95% confidence intervals (95% CIs) for the *P. falciparum* prevalence (%; x-axis) and ±2 standard deviations (±2 SD) for the number of unique upsA and non-upsA DBLα types (y-axis). The 95% CIs for *P. falciparum* prevalence (%) were calculated using the Wald interval. The ±2 SD for the number of unique upsA and non-upsA DBLα types was calculated based on a bootstrap approach. We resampled 10,000 replicates from the original population-level distribution with replacement. Each resampled replicate has the same size as the original sample. We then derive the standard deviation (SD) based on the distribution of the resampled replicates.

2015, coincident with the increase in *P. falciparum* prevalence and census population size (***Figures 4 and 5***).

Given this reduction in DBLα type richness following the IRS intervention and subsequent rebound in 2017, we wanted to explore whether the loss of richness was influenced by the frequency of individual DBLα types in the parasite population within and among surveys. To answer this, we defined the relative frequency of individual DBLα types in all isolates in each survey (***Figure 6***, ***Figure 6—figure supplement 1***). We discovered that individual upsA and non-upsA DBLα types were not all at equal frequencies within a survey and among surveys. They could be classified as frequent (i.e. observed in 11–20 or >20 isolates), less frequent (i.e. observed in 2–10 isolates), or only seen once, at baseline in 2012 (***Figure 6AB***). In 2014 and 2015, following IRS, there was a significant increase in the proportion of upsA and non-upsA DBLα types in the lower frequency categories (p-value<0.001, Mann-Whitney U test), with all DBLα types becoming rarer in the population (***Figure 6CD***). This change can be attributed to the removal of *P. falciparum var* repertoires (***Figure 4***) with associated loss of upsA and non-upsA DBLα type richness (***Figure 5***), which disproportionally affected those DBLα types seen once. This shift to all DBLα types becoming rarer following IRS changed in 2017, where the proportion of DBLα types in the more frequent categories (i.e. 2–10, 11–20, or >20 isolates) significantly increased while the proportion seen once decreased (p-values<0.001, Mann-Whitney U tests) (***Figure 6C and D***). Data in ***Figure 6A–D*** pointed to a differential effect of the IRS intervention and subsequent rebound on the less frequent upsA and non-upsA DBLα types vs. those that were classified as frequent, where those DBLα types that were most frequent persisted longitudinally.

To explore this observation further, we restricted the longitudinal analysis to those DBLα types from the baseline survey in 2012 (pre-IRS). We compared the probability of survival for the DBLα types identified at baseline in 2012 and found that the upsA DBLα types persisted significantly longer in the population relative to the non-upsA DBLα types (p<0.001, log-rank test), despite the IRS intervention.

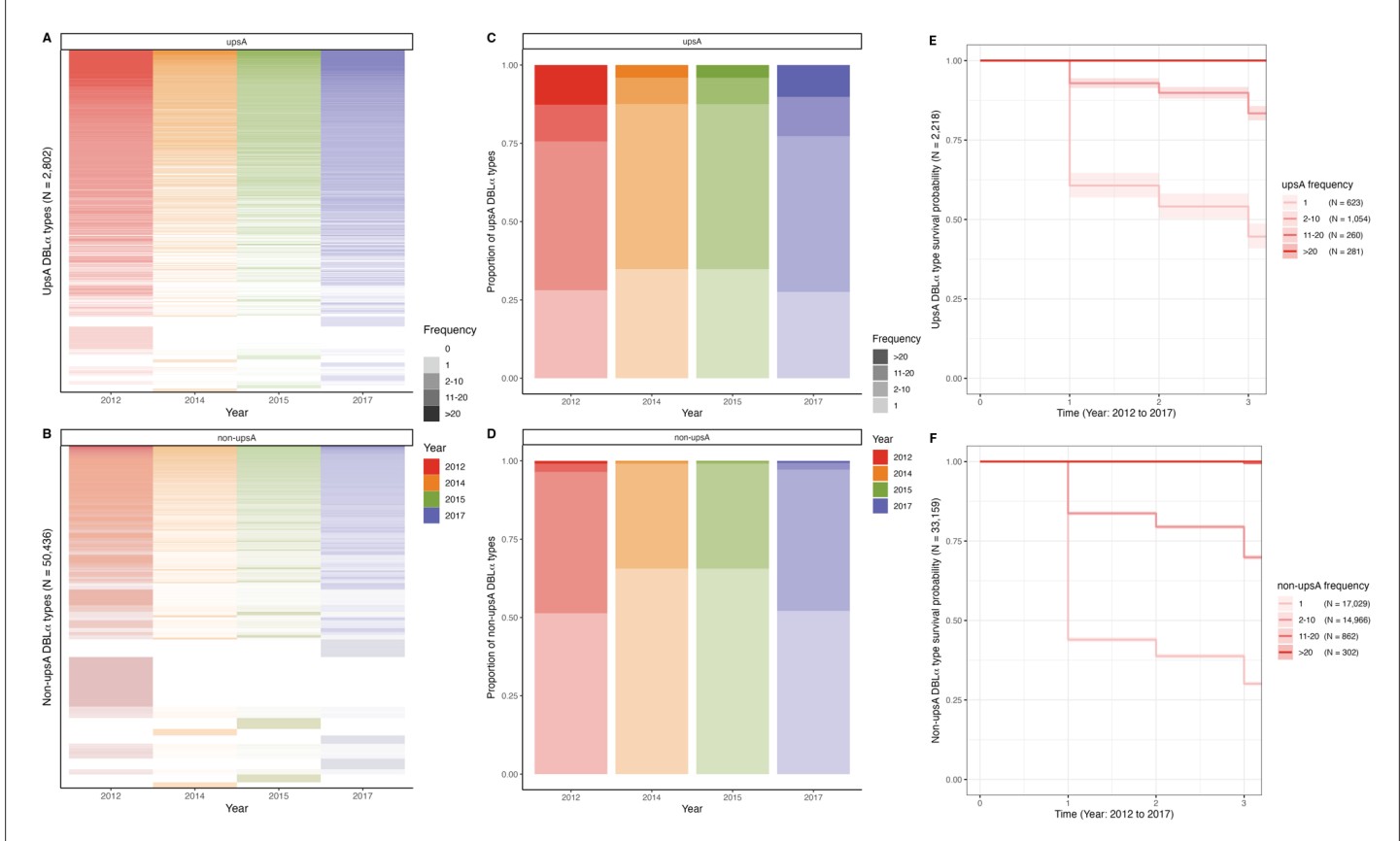

**Figure 6.** UpsA and non-upsA DBLα type frequencies and survival in 2012 (pre-indoor residual spraying [IRS], red), 2014 (during IRS, orange), 2015 (post-IRS, green), and 2017 (seasonal malaria chemoprevention [SMC], purple). Heatmaps showing the patterns of diversity for the (**A**) upsA and (**B**) non-upsA DBLα types. The columns represent all the upsA and non-upsA DBLα types observed in the four surveys, and the rows represent each of the 2802 upsA DBLα types and the 50,436 non-upsA DBLα types (*Appendix 1—table 3*). White rows are used to denote the absence of a specific DBLα type, while the presence of a DBLα type is indicated in colour and further categorised (colour gradations) based on the frequency or the number of times (i.e. number of isolates) a DBLα type was observed in each survey (frequency categories: 1, 2–10, 11–20, >20 isolates). Note the frequency category cut-offs were chosen based on the frequency distributions in *Figure 6—figure supplement 1*. The proportions of (**C**) upsA and (**D**) non-upsA DBLα types in each survey based on the number of times (i.e. number of isolates) they were observed in each survey. Kaplan-Meier survival curves for the (**E**) upsA and (**F**) non-upsA DBLα types across time (2012–2017) categorised based on their frequency at baseline in 2012 (pre-IRS, red). The coloured shaded areas represent the upper and lower limits of the 95% confidence intervals (95% CIs), with the number (**N**) of upsA and non-upsA DBLα types in each frequency category provided in parenthesis. These survival curves include only those upsA (N=2218) and non-upsA (N=33,159) DBLα types that were seen at baseline in 2012 (pre-IRS) as indicated in red (*Appendix 1—table 3*). The x-axis indicates time where time '0' denotes 2012 (pre-IRS), '1' denotes 2014 (during IRS), '2' denotes 2015 (post-IRS), and finally '3' denotes 2017 (SMC). *Note:* In the survival curves, the 11–20 and >20 frequency categories for both the (**E**) upsA and (**F**) non-upsA DBLα types overlap in the figure.

The online version of this article includes the following figure supplement(s) for figure 6:

**Figure supplement 1.** Frequency distributions for the (**A**) upsA and (**B**) non-upsA DBLα types in 2012 (pre-indoor residual spraying [IRS], red), 2014 (during IRS, orange), 2015 (post-IRS, green), and 2017 (seasonal malaria chemoprevention [SMC], purple).

The simple explanation being that although the upsA DBLα types had lower richness (*Figure 5*), a larger proportion was classified as frequent, indicating that multiple copies existed in the population compared to the non-upsA DBLα types (*Figure 6C and D*). Furthermore, when we examined survival using the frequency categories, the upsA and non-upsA DBLα types that were observed at multiple study time points (i.e. 2012, 2014, 2015, and 2017), albeit in different isolate repertoires, were those that were most frequent (i.e. observed in 11–20 and >20 isolates) in the population at baseline in 2012 (*Figure 6E and F*). As expected, the DBLα types that were only observed once in 2012 were significantly less likely to be seen longitudinally (p-value<0.001, log-rank test) (*Figure 6E and F*). These differential changes in DBLα type richness with respect to rare vs. frequent DBLα types are a

consequence of changes in census population size with interventions (*Figure 4*) where each isolate repertoire is composed of many rare DBLα types as defined by PTS (*Figure 2*).

## Discussion

*P. falciparum* populations in high-transmission endemic areas in SSA are characterised by extensive diversity, high rates of recombination, as well as frequent multiclonal infections. Here, we defined census population size of *P. falciparum* to understand total parasite diversity in a human population and explore the age-specific efficacy of malaria interventions to reduce this metric in such areas as typified by Bongo, Ghana. Census population size proved more informative than parasite prevalence alone because it captures 'within' host parasite population size as MOI, rather than using the infected host per se as a unit of population size. Whilst the concept of census population size is agnostic of how you measure MOI, the extensive DBLα isolate repertoire diversity presented makes a strong case for fingerprinting parasite isolates by *var*coding in high transmission. This is opposed to looking at allelic diversity of a single copy antigen gene, such as *csp*.

Census population size is a total enumeration or count of infections in a given population sample and over a given time period in an ecological sense, distinct from the formal effective population size ($N_e$) used in population genetics (*Charlesworth, 2009*). Given the low overlap between *var* repertoires of parasites observed in monoclonal infections (MOI=1), the census population size calculated in Bongo, Ghana, translates to a diversity of strains or repertoires. The distinction of census population size in terms of infection counts and effective population size from population genetics has been made before for pathogens, including the seasonal influenza virus and the measles virus (*Bedford et al., 2011*) it is also a distinction made in the ecological literature for non-pathogen populations (*Palstra and Fraser, 2012*). The census population size of a given population sample depends, of course, on sample size and was used here for comparisons across time of samples of the same depth (i.e. ~2000 individuals). There is, however, a simple map between census population size and mean MOI, as one can simply divide or multiply by the sample size, respectively, to convert between the two quantities (*Appendix 1—table 2*). Therefore, one can extrapolate from the census population size of a given population sample to that of the whole population of local hosts in a given area to compare across studies that differ in sampling depth and/or spatial extent. What is needed for this extrapolation is a stable mean MOI relative to the sample size or sampling depth, which is indeed the case in this study (*Appendix 1—figure 1*) and can be easily checked in other studies. Given the typical duration of infection, we expect our population size to be representative of a per-generation measure.

By *var*coding, we identified a very large census parasite population size in a relatively small human population of ~2000 individuals at baseline and captured age-specific changes in this metric in response to sequential malaria control interventions. IRS reduced the MOI$_{var}$ parasite population size substantially with the greatest reductions (85%) seen in the younger children (1–5 years). More than two years after the cessation of IRS, the rebound in 2017 was rapid in all age groups, except for the younger children (1–5 years) where those up to 59 months were targeted by SMC. Population sizes in adolescents (11–20 years) and adults (>20 years) showed they carried more infections in 2017 than at baseline in 2012. This is indicative of a loss of immunity during IRS which may relate to the observed loss of *var* richness, especially the many rare types. This warrants further investigation of changes in variant-specific immunity. During and following the IRS and SMC interventions, *var* diversity remained high and *var* repertoire overlap remained low, reflecting characteristic properties of high transmission and demonstrating the overall resilience of the species to survive significant short-term perturbations. Combining interventions and targeting older age groups or the whole community with chemoprevention would no doubt have a much greater impact on reducing the diversity of the reservoir of infection.

What was striking about the Bongo study was the speed with which rebound in MOI$_{var}$ per person and census population size occurred, once the short-term IRS was discontinued. We looked for a potential explanation in our genetic data. PTS and population frequency data showed that many of the DBLα types occurred in multiple repertoires or genomes. This enabled the survival of these more frequent DBLα types through the interventions, facilitating rebound by maintenance of this diversity. The other notable population genetic result of our study was the failure to increase similarity (or relatedness by state) of *var* repertoires by reducing transmission by >90% via IRS. From a baseline of a very large population size with very low overlap in repertoires, you need outcrossing to create relatedness. However, this was less likely to happen due to reduced transmission as a result of IRS. Rebound,

with associated increases in transmission, led to a small increase in *var* repertoire similarity. This is the opposite to what has been observed in areas of lower transmission under intense malaria control where the intensity of interventions led to increased genome similarity, as assessed by identity-by-descent (IBD), from a starting point of much lower genome diversity and greater relatedness (*Daniels et al., 2015*).

Our molecular approach to measure population size has been to sum $MOI_{var}$ in individual hosts with microscopically detectable infections. Like any diagnostic method, there are limits to sensitivity and specificity, which can be more or less tolerated dependent upon the purpose of the study. Here, we have looked at relative changes in population size with sequential interventions using an interrupted time-series study design and observed changes by measuring $MOI_{var}$. We have accounted for missing DBLα type data where complete *var* repertoires may not have been sequenced using a Bayesian method based on empirical knowledge of the measurement error. This approach has a conceptual relation to the Bayesian approach by *Johnson and Larremore, 2022*, to estimate complete repertoire size of, and overlap between, monoclonal infections from incomplete sampling of DBLα types. Our measurement of population size based on $MOI_{var}$ will be subject to other sampling errors which may in the end be more significant (discussed in detail in *Labbé et al., 2023*). For example, low parasitaemia typical of asymptomatic infections, small blood volumes, clinical status, and/or within host dynamics, including synchronicity, will all create sampling problems, but these are common to all measures of MOI.

Previously, we have drawn attention to the potential underestimation of the number of DBLα types of related parasites generated by a cross, when using *var* genotyping (*Labbé et al., 2023*). Such related parasites must be created frequently in high transmission due to extensive outcrossing (*Babiker et al., 1994*; *Paul et al., 1995*). Single clone genomics experiments using biallelic SNPs from whole genome sequencing data have also detected related parasites using IBD in clinical infections from humans in a high-transmission area of Malawi (*Nkhoma et al., 2020*). Here, we have analysed low- to moderate-density, chronic, asymptomatic infections (see *Appendix 1—table 1*) under strong immune selection in semi-immune hosts whom we have shown select against parasites with high PTS scores consistent with relatedness by descent (*Day et al., 2017*; *He et al., 2018*; *Ruybal-Pesántez et al., 2022*). When considering the importance of possible exclusion of parasites related by descent, the only sure way to detect such parasites in high transmission is by single cell genomics, a methodology of limited application to malaria surveillance due to practicality and cost of scale up. Again, the error from failure to sample infections related by descent must be weighed up against the issues of under-sampling as described above.

The Bayesian approach to the *var*coding method relies on the low or limiting similarity of *var* repertoires infecting individual human hosts. As such, it would appear to break down as the *var* repertoire overlap moves away from extremely low, and therefore, for locations with lower transmission intensity. Interestingly, this is not the case in the numerical simulations of *Labbé et al., 2023*, for a gradient of three transmission intensities, from high to low, with the original *var*coding method performing well across the gradient. This robustness of the method may arise from a nonlinear and fast transition from low to high overlap that is accompanied by MOI changing rapidly from primarily multiclonal (MOI>1) to monoclonal (MOI=1) infections. This matter needs to be investigated further in the future, including ways to extend the Bayesian approach to explicitly include the distribution of *var* repertoire overlap.

In summary, our findings provide parasite population insights into why rebound is the inevitable consequence of such short-term IRS interventions unless you simultaneously target the highly diverse, long-lived parasite population in humans, not just children <5 years by SMC. Of potential translational significance for malaria molecular surveillance, we identify new metrics, especially $MOI_{var}$ and census population size, as well as *var* frequency category, informative to monitor and evaluate interventions in high-transmission areas with multiclonal infections and high rates of outcrossing. Such metrics could be used longitudinally to detect incremental gains of transmission-reducing interventions, including IRS, long-lasting insecticidal nets (LLINs), and vaccines to perturb the high-transmission characteristics of the parasite population in humans in high-burden countries in SSA.

# Materials and methods

**Key resources table**

| Reagent type (species) or resource | Designation | Source or reference | Identifiers | Additional information |
|---|---|---|---|---|
| Commercial assay or kit | QIAamp DNA mini kit | QIAGEN | Cat #: 51306 | With modifications as described in *Tiedje et al., 2017* |
| Sequence-based reagent | dNTP mix | Promega | Cat #: U1511, U1515 | See Appendix 1 |
| Sequence-based reagent | GoTaq G2 Flexi DNA polymerase | Promega | Cat #: M7805 | See Appendix 1 |
| Sequence-based reagent | DBLaAF-MID | *Rask et al., 2016* | Forward PCR primers | See Appendix 1 |
| Sequence-based reagent | DBLaBR-MID | *Rask et al., 2016* | Reverse PCR primers | See Appendix 1 |
| Other | AMPure XP Beads for DNA Cleanup | Beckman Coulter | Cat #: A63880, A63881 | See Appendix 1 |
| Commercial assay or kit | Quant-iT PicoGreen dsDNA Assay Kit | Invitrogen | P11496 | See Appendix 1 |
| Commercial assay or kit | KAPA HiFi Taq HotStart Ready Mix | Roche | Cat #: KK2601 | See Appendix 1 |
| Software, algorithm | R 4.3.1 | *R Development Core Team, 2018* | | |

## Human subject ethical approval

The study was reviewed/approved by the ethics committees at the Navrongo Health Research Centre Ghana (NHRC IRB-131), Noguchi Memorial Institute for Medical Research, Ghana (NMIMR-IRB CPN 089/11-12; NMIMR-IRB CPN 066/20-21), The University of Chicago, United States (IRB14-1495; IRB19-0760; IRB21-0417), and New York University, United States (IRB-FY2024-8572), and The University of Melbourne, Australia (Project IDs 13433, 31586, 21649). Individual informed consent was obtained in the local language (i.e. Gurene) from each participant enrolled by signature or thumbprint, accompanied by the signature of an independent witness. For children <18 years of age, a parent or guardian provided consent. In addition, all children between the ages of 12 and 17 provided assent. Details on the study area, study population, inclusion/exclusion criteria, and data collection procedures have been previously described (*Tiedje et al., 2022*, *Tiedje et al., 2017*).

## Study design and sample collection

Using an interrupted time-series study design, four age-stratified cross-sectional surveys of ~2000 participants per survey were undertaken to investigate the impacts of IRS and SMC in combination with LLINs impregnated with pyrethroids under operational conditions on the asymptomatic *P. falciparum* reservoir from two proximal catchment areas in Bongo District, Ghana (hereinafter referred to collectively as 'Bongo'; *Tiedje et al., 2022*, *Tiedje et al., 2017*; *Appendix 1—table 1*). Bongo District, located in the Upper East Region, is categorised as high, seasonal malaria transmission based on the World Health Organization's (WHO) 'A Framework for Malaria Elimination' (WHO/*WHO/GMP, 2017*) where *P. falciparum* prevalence was ≥35% at baseline in 2012 (*Tiedje et al., 2022*, *Tiedje et al., 2017*). These ~2000 participants of all ages (1–97 years) represent ~15% of the total population that resides in these two catchment areas in Bongo (*Tiedje et al., 2017*). The four cross-sectional surveys were completed at the end of the wet season (i.e. high-transmission season) and the study can be separated into four distinct study time points: (1) October 2012 (Survey 1) prior to the IRS and SMC (i.e. baseline), (2) October 2014 (Survey 2) two months after the second round of IRS (Actellic 50EC), (3) October 2015 (Survey 3) seven months after the third round of IRS using a long-acting non-pyrethroid insecticide (Actellic 300CS) (*Gogue et al., 2020*; *US President's Malaria Initiative Africa IRS (AIRS) Project, 2016*), and finally (4) October 2017 (Survey 4) 32 months after the discontinuation of IRS, but during the deployment of SMC to all children 3–59 months (i.e. <5 years) (*Figure 1A*). LLINs (i.e. PermaNet 2.0, Olyset, or DawaPlus 2.0) were mass distributed in Bongo District by the National Malaria Elimination Programme (NMEP)/Ghana Health Service (GHS) between 2010 and 2012 and again in 2016 following the discontinuation of IRS (*Gogue et al., 2020*; *Tiedje et al., 2022*; *US*

*Agency for International Development (USAID) Global Health Supply Chain Program, 2020*). In addition, to maintain high coverage of LLINs between these campaigns, continuous distribution was undertaken using routine services (i.e. antenatal clinics, school distributions, immunisation visits, etc.) (*Tiedje et al., 2022*). Over the course of this study, self-reported LLIN usage from the previous night remained high across all age groups: from 89.1% in 2012 (pre-IRS), 83.5% in 2014 (during IRS), 90.6% in 2015 (post-IRS), to 96.8% in 2017 (SMC). Details on the study area, study population, and data collection procedures have been previously described (*Tiedje et al., 2022*, *Tiedje et al., 2017*).

## Details of the IRS and SMC interventions

Starting in 2013, the AngloGold Ashanti Malaria Control Programme (AGAMal) in a public-private partnership with the Global Fund scaled up IRS across all of the Upper East Region of northern Ghana (*Gogue et al., 2020*). As part of this initiative, three rounds of IRS with organophosphate formulations (i.e. non-pyrethroid) were rolled out prior to the start of the wet season between 2013 and 2015 (*Figure 1A*) in Bongo District (*Tiedje et al., 2022*). Based on AGAMal's operational reports, IRS coverage in Bongo District was 91.8% in Round 1, 95.6% in Round 2, and finally 96.6% in Round 3 (AGAMal, personal communication). To monitor the impact of the IRS on the local vector population, monthly entomological collections were undertaken between February 2013 and September 2015 (*Tiedje et al., 2022*). Using these surveys, we observed that the monthly EIR (infective bites/person/month [ib/p/m]), a measure of local transmission intensity, declined by >90% at the peak of the wet season between August 2013 (pre-IRS) (EIR = 5.3 ib/p/m) and August 2015 (post-IRS) (EIR = 0.4 ib/p/m) (*Tiedje et al., 2022*). Following the IRS, SMC was rolled out in the Upper East Region by the NMEP/GHS starting in 2016 (*Gogue et al., 2020*; *Figure 1A*). SMC is the intermittent administration of a curative dose of an antimalarial to children between the ages of 3–59 months (i.e. <5 years) (*WHO, 2012*). Like other countries, the SMC drug combination of choice in Ghana is sulfadoxine-pyrimethamine plus amodiaquine, which is administered at monthly intervals (i.e. ~28–30 days apart) during the high-transmission season (i.e. wet season) (*WHO, 2012*). The goal of this age-targeted intervention is to both clear current infections and prevent malarial illness by maintaining a therapeutic concentration of an antimalarial in the blood over the period of greatest risk (i.e. high-transmission season). Reported SMC coverage in Bongo District was 92.6% in 2016 (two cycles between August and September 2016) and 94.6% in 2017 (four cycles between July and October 2017) (*Gogue et al., 2020*; NMEP/GHS, personal communication).

## *Var* genotyping and sequence analysis

Genomic DNA was extracted from the dried blood spots for all participants with a confirmed microscopic asymptomatic *P. falciparum* infection (i.e. isolate) (2572 isolates, *Appendix 1—tables 2 and 3*) using the QIAamp DNA mini kit (QIAGEN, USA) with modifications as previously described (*Tiedje et al., 2017*). For *var* genotyping or *varcoding*, the sequence region within the *var* genes encoding the DBLα domains of PfEMP1 was amplified using a single-step PCR, pooled, and sequenced on an Illumina platform using the MiSeq Reagent Kit v3 (600 cycle; 2×300 bp paired-end) (see Appendix 1, *Appendix 1—figure 2*). The raw sequence data was then processed using our previously published customised bioinformatic pipelines (see Appendix 1, *Appendix 1—figure 3*). For additional information on the use of these bioinformatic pipelines, a detailed tutorial is available (; *Tiedje and Tan, 2025*).

DBLα sequencing data was obtained from 2397 *P. falciparum* isolates (93.2%) (*Appendix 1—table 3*). This genotyping success was acceptable given that we were working with low-density asymptomatic infections (*Appendix 1—table 1*). Using a cut-off of ≥20 DBLα types to ensure robust downstream analyses, DBLα sequencing data was obtained from 2099 *P. falciparum* isolates (81.6%) with a total of 289,049 DBLα sequences and 53,238 unique DBLα types being identified in the study population (*Appendix 1—table 3*). The median *P. falciparum* density was ~4 times higher for isolates with ≥20 DBLα types compared to those that gave no or <20 DBLα types (520 [IQR: 200–1880] parasites/μL vs. 120 [IQR: 40–200] parasites/μL, respectively).

## DBLα type diversity

We monitored the impacts of the sequential interventions (i.e. IRS and SMC) on diversity by measuring changes in the population genetics of DBLα types at the population level (i.e. *P. falciparum* reservoir).

Diversity was monitored using two measures, DBLα type richness and DBLα type frequency. Richness was defined as the number of unique DBLα types observed (i.e. DBLα type pool size) in each survey or study time point (i.e. 2012, 2014, 2015, and 2017). DBLα type richness, however, does not provide any information about the relative frequencies of the different DBLα types in the population, as they are all weighted equally whether they are observed once or more frequently (e.g. observed in >20 isolates per survey). To further examine the impacts of the interventions on DBLα type diversity, we also assessed the frequency of each unique DBLα type in 2012, 2014, 2015, and 2017. Here, we defined DBLα type frequency as the number of times (i.e. number of isolates) a DBLα type was observed in each survey. Both upsA and non-upsA DBLα type diversities were measured due to their different biological features, chromosomal positions (i.e. subtelomeric regions vs. internal or central regions), as well as population genetics (*Falk et al., 2009*; *Gardner et al., 2002*; *Jensen et al., 2004*; *Kaestli et al., 2006*; *Kalmbach et al., 2010*; *Kraemer et al., 2007*; *Kraemer and Smith, 2006*; *Kyriacou et al., 2006*; *Lavstsen et al., 2003*; *Normark et al., 2007*; *Rottmann et al., 2006*; *Warimwe et al., 2012*; *Warimwe et al., 2009*; *Zhang and Deitsch, 2022*). The proportions of upsA and non-upsA *var* genes in a repertoire or single genome have been defined as ~15–20% and ~80–85%, respectively, based on whole-genome sequencing (*Rask et al., 2010*). The upsA and non-upsA DBLα type proportions were partitioned as expected in our analyses, with the median proportions at the repertoire level being comparable in 2012 (19% upsA and 81% non-upsA), 2014 (22% upsA and 78% non-upsA), 2015 (21% upsA and 79% non-upsA), and 2017 (20% upsA and 80% non-upsA).

## Repertoire similarity as defined by pairwise type sharing

To estimate genetic similarity between the DBLα repertoires (i.e. unique DBLα types identified in each isolate) identified from two isolates, pairwise type sharing (PTS) was calculated between all pairs of isolates in each survey as previously described (*Barry et al., 2007*). PTS, analogous to the Sørensen index, is a similarity statistic to evaluate the proportion of DBLα types shared between two isolate repertoires (i.e. DBLα repertoire overlap) and ranges from 0 (i.e. no DBLα repertoire overlap) to 1 (i.e. identical DBLα isolate repertoires), where <0.50 = unrelated, 0.5=recent recombinants/siblings, >0.5 = related, and 1=clones. PTS is a measure of identity-by-state used to assess repertoire similarity between isolates and is not used to infer inheritance from a recent common ancestor (i.e. IBD) (*Speed and Balding, 2015*).

## DBLα isolate repertoire size

For this study, we have exploited the unique population structure of non-overlapping DBLα isolate repertoires to estimate isolate $MOI_{var}$. To calculate $MOI_{var}$, the non-upsA DBLα types were chosen since not only are they more diverse and less conserved between isolate repertoires (i.e. low median $PTS_{non-upsA} \leq 0.020$) compared to the upsA DBLα types, but they have also been shown to have a more specific 1-to-1 relationship with a single *var* gene than upsA (*Tan et al., 2023*). The low to non-existent overlap of repertoires enables an estimation of MOI that relies on the number of non-upsA DBLα types sequenced from an individual's isolate (*Ruybal-Pesántez et al., 2022*; *Tiedje et al., 2022*). A constant repertoire size or number of DBLα types in a parasite genome can be used to convert the number of types sequenced in an isolate to estimate MOI (*Ruybal-Pesántez et al., 2022*; *Tiedje et al., 2022*). This approach, however, neglects the measurement error in this size introduced by targeted PCR and amplicon sequencing of *var* genes in an isolate.

## Bayesian estimation of $MOI_{var}$ and associated census population size

Here, we extend the method to a Bayesian formulation and estimate the posterior distribution for each sampled individual for the probability of different MOI values. From individual posterior distributions, we can then obtain the estimated MOI frequency distribution for the population as a whole. The two pieces of information required for our approach are the measurement error and the prior distribution of MOI. The measurement error is simply the repertoire size distribution, i.e.the distribution of the number of non-upsA DBLα types sequenced given MOI=1, which is empirically available (*Appendix 1—figure 4*; *Labbé et al., 2023*). We refer to it as P(s | MOI = 1) where s here denotes repertoire size. More generally, when MOI≥1, s denotes the number of non-upsA DBLα types sequenced, which corresponds to the repertoire or isolate size. The prior distribution of MOI refers to the belief we have for what the actual MOI distribution might look like at the population level before

empirical evidence is taken into consideration. For example, the prior distribution of MOI is likely to centre around a higher value in high-transmission endemic areas than in low-transmission ones.

We can obtain P(s | MOI = m) from the serial convolution of the repertoire size distribution P(s | MOI = 1) and P(s | MOI = m – 1). Starting with the repertoire size distribution given a single infection, we can derive P(s | MOI = m) for m equal to 2,3,…, up until a maximum value of 20 (empirically determined), as follows:

$$P\left(s|MOI = m\right) = \sum_{x=L}^{U} P\left(x|MOI = 1\right) \times P\left(s - x|MOI = m - 1\right) \tag{1.1}$$

where L and U are the lower and upper limit for the repertoire size, 10 and 45, respectively, from the empirical repertoire size distribution (*Appendix 1—figure 4*; *Labbé et al., 2023*).

For simplicity, we begin with a uniform prior. We use Bayes' rule to derive a posterior distribution of MOI given a certain number of non-upsA DBLα types sequenced from an individual:

$$P\left(MOI = j|s\right) = \frac{P\left(s|MOI = j\right) \times P\left(MOI = j\right)}{\sum_{i=1}^{k} P\left(s|MOI = i\right) \times P\left(MOI = i\right)} \tag{1.2}$$

where k is the maximum value of MOI, here 20, as empirically determined.

To obtain the MOI distribution at the population level, we could either simply pool the maximum *a posteriori* MOI estimate for each sampled individual, or use a technique called mixture distribution. For the latter, we weighed each posterior MOI distribution for each sampled individual equally and summed over all posterior distributions at the individual level to derive the MOI distribution at the population level:

$$f\left(MOI = m\right) = \sum_{i=1}^{n} \frac{1}{n} P\left(MOI = m|s_i\right) \tag{1.3}$$

where n is the number of sampled individuals. These two approaches gave similar results for our empirical survey data as determined by the Kolmogorov-Smirnov test. The obtained distance statistic is close to 0 and the corresponding p-value is non-significant across all surveys, indicating that the two estimates were drawn from the same distribution (*Appendix 1—tables 4 and 5*). Additionally, the difference between the mean MOI values at the population level obtained from the two approaches is small (*Appendix 1—tables 4 and 5*). Given this similarity, we present the results based on pooling the maximum *a posteriori* MOI estimates for each sampled individual in the main text and include the results based on mixture distribution in Appendix 1. Note that we focused on individuals who had confirmed microscopic asymptomatic *P. falciparum* infections for our MOI estimation.

To examine alternative priors, we considered empirical MOI distributions described in the literature including the Poisson, hyper-Poisson, and negative binomial distributions (*Dietz, 1988*; *Henry, 2020*). The hyper-Poisson and negative binomial distributions can capture the overdispersion seen in the empirical distribution of MOI for certain areas and caused by factors such as heterogeneous biting. We therefore focused on a negative binomial distribution and investigated changing its parameters to generate priors with different means spanning a wide range of MOI values (mean MOI within [~1.5, ~6.7]), including those seen in high-transmission endemic areas. A uniform prior and a zero-truncated negative binomial distribution with parameters within the range typical of high-transmission endemic regions (higher mean MOI, e.g., ~4.3 vs. ~6.7, with tails for higher MOI values in the range of 10–20) produce similar MOI estimates at the population level (*Appendix 1—tables 6 and 7*). However, when setting the parameter range of the zero-truncated negative binomial to be representative of those in low-transmission endemic regions where the empirical MOI distribution centres around monoclonal infections with the majority of MOIs being 1 or 2 (mean MOI≈1.5, no tail at higher MOI values), the final population-level MOI distribution does deviate more from that based on the aforementioned prior and parameter choices (*Appendix 1—tables 6 and 7*). The final individual- and population-level MOI estimates are not sensitive to the specifics of the prior MOI distribution as long as this distribution captures the tail for higher MOI values with above-zero probability. The obtained Kolmogorov-Smirnov test distance statistics and their corresponding p-values, the Pearson correlation tests and

their corresponding p-values, as well as the difference in mean MOI values, for the comparison of the MOI estimates obtained with the different priors are included in *Appendix 1—tables 6 and 7*. Given these comparisons, we provide in our analyses the estimated population MOI distribution using a uniform prior.

## Adjusting for differences in sampling depth across age groups and surveys in the census population size calculation

In our interrupted time-series study design, we sampled the Bongo population by age group to assess age-specific effects of the sequential interventions (i.e. IRS and SMC). Sampling depth by age group was consistent across the four cross-sectional surveys (*Appendix 1—table 1*), except for the younger children (1–5 years) in 2014. To account for variation in sampling depth across age groups and surveys, we performed subsampling at the level of the minimum number of individuals in each age group across all surveys. We calculated the total number of *var* repertoires from a subsample of this number of individuals within each age group for each survey. This approach ensures consistent sample sizes within each age group across all surveys. We summed the *var* repertoires across age groups to obtain the total *var* repertoire count for each survey. To calculate the mean and 95% confidence interval (95% CIs) for the number of *var* repertoires (i.e. census population size), we repeated the subsampling procedure 10,000 times and derived these quantities from the distribution of these repeated subsampling replicates.

## Statistical analysis

We used the R v4.3.1 for all data analyses with the collection of R packages in *tidyverse* being used for data curation along with *base, stats, gtsummary,* and *epiR* for the statistical analyses (*R Development Core Team, 2018*; *Sjoberg et al., 2021*; *Stevenson, 2020*; *Wickham et al., 2019*). Continuous variables are presented as medians with IQRs, and discrete variables are presented using the observed or calculated values with the 95% confidence interval (95% CIs) or ±2 standard deviations (±2 SD). Kaplan-Meier survival curves were generated for the time (i.e. number of surveys) to first event (i.e. when the DBLα type was no longer observed/detected) comparing the upsA and non-upsA DBLα types; p-values were determined using the log-rank test using the R packages *survival* and *survminer* (*Kassambara et al., 2021*; *Therneau, 2023*). The time interval to first event considered for all survival curves was the number of surveys or year (i.e. 2012, 2014, 2015, and 2017) that each DBLα type was observed and only includes those upsA (N=2218) and non-upsA (N=33,159) DBLα types that were seen at baseline in 2012 (i.e. those DBLα types observed prior to the IRS intervention) (*Appendix 1—table 3*).

## Benefit-sharing statement

A research collaboration was developed with scientists from Ghana based at the Navrongo Health Research Centre and the Noguchi Memorial Institute for Medical Research. All collaborators are included as co-authors, and the relevant results from the research have been shared with participants, key stakeholders, and the local community (i.e. Paramount Chief of Bongo District, divisional Chiefs, Queen Mothers, and community members), the Bongo District and Upper East Regional Health Directorates, as well as Ghana National Malaria Elimination Programme. Before this research was undertaken, informed consent was sought and obtained from the key stakeholders, traditional leadership, and the local community in Bongo District. In addition, members of the local community were trained as field workers and were directly involved in liaising with the community and in the collection of the study data. The contribution of these individuals to this research is described in the Acknowledgements. This research addresses a priority concern regarding malaria control and the impact of interventions. These concerns are relevant to both the local community in Bongo District and the National Malaria Elimination Programme in Ghana.

## Acknowledgements

We wish to thank the participants, communities, traditional leadership, and the Ghana Health Service in Bongo District, Ghana, for their willingness to participate in this study. We would like to thank the field teams in Bongo for their technical assistance in the field, as well as the laboratory personnel at

the Navrongo Health Research Centre for their expertise and for undertaking the sample collections and parasitological assessments.

## Additional information

### Funding

| Funder | Grant reference number | Author |
| --- | --- | --- |
| Fogarty International Center | R01-TW009670 | Kwadwo A Koram<br>Mercedes Pascual<br>Karen P Day |
| National Institute of Allergy and Infectious Diseases | R01-AI149779 | Abraham R Oduro<br>Kwadwo A Koram<br>Mercedes Pascual<br>Karen P Day |

The funders had no role in study design, data collection and interpretation, or the decision to submit the work for publication.

### Author contributions

Kathryn E Tiedje, Data curation, Formal analysis, Validation, Investigation, Visualization, Methodology, Writing – original draft, Project administration, Processed, cleaned, and curated the datasets for analysis, Analysed the data, Processed the samples and performed the genotyping experiments, Provided input on the field study, Coordinated the field studies; Qi Zhan, Software, Formal analysis, Validation, Investigation, Visualization, Methodology, Writing – original draft, Analysed the data, Developed and implemented the Bayesian estimations; Shazia Ruybal-Pésantez, Data curation, Formal analysis, Investigation, Writing – review and editing, Processed, cleaned, and curated the datasets for analysis, Contributed to the data analyses, Processed the samples and performed the genotyping experiments; Gerry Tonkin-Hill, Validation, Methodology, Writing – review and editing, Developed and validated the bioinformatic pipelines; Qixin He, Formal analysis, Methodology, Writing – review and editing, Contributed to the data analyses; Mun Hua Tan, Formal analysis, Writing – review and editing, Contributed to the data analyses; Dionne C Argyropoulos, Data curation, Writing – review and editing, Processed, cleaned, and curated the datasets for analysis; Samantha Deed, Investigation, Writing – review and editing, Processed the samples and performed the genotyping experiments; Anita Ghansah, Methodology, Writing – review and editing, Provided input on the field study; Oscar Bangre, Investigation, Project administration, Writing – review and editing, Coordinated the field studies; Abraham R Oduro, Resources, Funding acquisition, Methodology, Project administration, Provided input on the field study; Kwadwo A Koram, Funding acquisition, Methodology, Writing – review and editing, Conceived and designed the field study; Mercedes Pascual, Conceptualization, Resources, Software, Formal analysis, Supervision, Funding acquisition, Methodology, Writing – original draft, Project administration, Conceptualized the study of population size using MOIvar, Analysed the data; Karen P Day, Conceptualization, Resources, Formal analysis, Supervision, Funding acquisition, Methodology, Writing – original draft, Project administration, Conceptualized the study of population size using MOIvar, Conceived and designed the field study, Analysed the data

### Author ORCIDs

Kathryn E Tiedje 
Qi Zhan 
Shazia Ruybal-Pésantez 
Qixin He 
Mun Hua Tan 
Dionne C Argyropoulos 
Kwadwo A Koram 
Karen P Day 

## Ethics

Human subjects: The study was reviewed/approved by the ethics committees at the Navrongo Health Research Centre Ghana (NHRC IRB-131), Noguchi Memorial Institute for Medical Research, Ghana (NMIMR-IRB CPN 089/11-12; NMIMR-IRB CPN 066/20-21), The University of Chicago, United States (IRB14-1495; IRB19-0760; IRB21-0417), and New York University, United States (IRB-FY2024-8572), and The University of Melbourne, Australia (Project IDs 13433, 31586, 21649). Individual informed consent was obtained in the local language (i.e. Gurene) from each participant enrolled by signature or thumbprint, accompanied by the signature of an independent witness. For children < 18 years of age, a parent or guardian provided consent. In addition, all children between the ages of 12 and 17 provided assent.

Reviewer #2 (Public review): https://doi.org/10.7554/eLife.91411.4.sa1
Author response https://doi.org/10.7554/eLife.91411.4.sa2

---

# Additional files

## Supplementary files

MDAR checklist

## Data availability

The sequences utilized in this study are publicly available in GenBank under BioProject Number: PRJNA 396962. All data associated with this study, including de-identified individual participant data, are available in the manuscript, appendices, and on GitHub at https://github.com/UniMelb-Day-Lab/Census_Pop_Size_Pf_Ghana. Redistribution or reuse of these data requires proper attribution and prior approval. Researchers interested in further use of these data should contact the Malaria Reservoir Study Team, represented by the corresponding author, Prof. Karen Day (karen.day@unimelb.edu. au), to discuss how these data will be utilized for academic or research purposes and, if appropriate, to identify opportunities for collaboration. The PCR protocol and primer sequences are described in Appendix 1 and available on GitHub (https://github.com/UniMelb-Day-Lab/Pfalciparum_varD-BLalpha_PCR; *Tiedje, 2025*). All custom code is available in an open source GitHub repository: (1) DBL Cleaner pipeline is available at https://github.com/UniMelb-Day-Lab/DBLaCleaner (*Tan and Tiedje, 2023*); (2) clusterDBLalpha pipeline is available at https://github.com/Unimelb-Day-Lab/clusterDBLalpha (*Tonkin-Hill and Tiedje, 2017*); and the (3) classifyDBLalpha pipeline is available at https://github.com/Unimelb-Day-Lab/classifyDBLalpha (*Tonkin-Hill and Pesántez, 2019*). A dataset and tutorial to demo this custom code is available at https://github.com/UniMelb-Day-Lab/tutorialD-BLalpha (*Tiedje and Tan, 2025*). For additional information on the use of the Bayesian approach to estimate MOIvar please see https://github.com/qzhan321/Bayesian-formulation-varcoding-MOI-estimation (*Zhan, 2024*).

The following datasets were generated:

| Author(s) | Year | Dataset title | Dataset URL | Database and Identifier |
|---|---|---|---|---|
| Malaria Reservoir Study Team | 2017 | Bongo District Ghana Study (GHSurvey7) | https://www.ncbi.nlm.nih.gov/biosample/SAMN41081346 | NCBI BioSample, SAMN41081346 |
| Tiedje KE, Zhan Q | 2025 | Measuring changes in *Plasmodium falciparum* census population size in response to sequential malaria control interventions | https://github.com/UniMelb-Day-Lab/Census_Pop_Size_Pf_Ghana | GitHub, Census_Pop_Size_Pf_Ghana |

The following previously published dataset was used:

| Author(s) | Year | Dataset title | Dataset URL | Database and Identifier |
|---|---|---|---|---|
| Malaria Reservoir Study Team | 2016 | Bongo District Ghana Study (GHPilot-GHSurvey6) | https://www.ncbi.nlm.nih.gov/biosample/SAMN11606536 | NCBI BioSample, SAMN11606536 |

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

# Appendix 1

## Materials and methods

### *Var* genotyping and sequence analysis

For *var* genotyping, the sequence region within the *var* genes encoding the DBLα domain of PfEMP1 was amplified in a single-step PCR from genomic DNA using primers for multiplexed sequencing as reported in *Rask et al., 2016*. We coupled template-specific degenerate primer sequences targeting homology block 2 (forward primer: DBLaAF, 5'-GCACGMAGTTTYGC-3') and homology block 3 (reverse primer: DBLaBR, 5'-GCCCATTCSTCGAACCA- 3') (*Bull et al., 2005*; *Rask et al., 2010*; *Taylor et al., 2000*) with a GS FLX Titanium primer sequence. Each of the forward and reverse DBLα primers was barcoded with a unique 10 bp multiplex identifier (MID) tag published by *Roche, 2009*. For a complete list of the primer sequences, see https://github.com/UniMelb-Day-Lab/Pfalciparum_varDBLalpha_PCR (*Tiedje, 2025*). For additional details on the validation of these primers for amplification of sequences of the appropriate length (~477 bp) using *P. falciparum* reference strains (3D7, Dd2, and HB3), see *Rask et al., 2016*.

Each PCR reaction was prepared in a total volume of 40 µL consisting of 0.5x buffer, 2 mM of MgCl$_2$, 0.07 mM of dNTP mix (Promega), 0.375 µM of each primer (DBLαAF, DBLαBR), 3 units of GoTaq G2 Flexi DNA polymerase (Promega), and 2 µL of genomic DNA template. The PCR cycling conditions involved 95°C for 2 min, followed by 30 cycles of 95°C for 40 s, 49°C for 90 s, 65°C for 90 s, and a final extension step of 65°C for 10 min. Positive controls (laboratory genomic *P. falciparum* DNA) and a negative control (no template) were included for quality assurance. The PCR products were purified using the SPRI method (solid-phase reversible immobilisation) (AMPure XP Beads, Beckman Coulter). Purified PCR product concentrations were measured using the Quant-iT PicoGreen dsDNA kit as per the manufacturer's instructions (Invitrogen). We assayed fluorescence intensity using a Perkin-Elmer VICTOR X3 multilabel plate reader, with fluorescein excitation wavelength of ~480 nm and emission of ~520 nm wavelength. Amplicons were then pooled equimolarly with each pool consisting of up to 106 isolates, all with unique MID tags. For full details on the PCR protocol and primer sequences, see https://github.com/UniMelb-Day-Lab/Pfalciparum_varDBLalpha_PCR. The libraries were then prepared using the KAPA HiFi HotStart Ready Mix (Roche) and sequenced on an Illumina platform using the MiSeq Reagent Kit v3 (600 cycle; 2×300 bp paired-end) (New York University Genome Technology Center, New York, NY, USA; Australian Genome Research Facility, Melbourne, Australia) (*Appendix 1—figure 2*).

The raw sequence data was then cleaned using our published customised bioinformatic pipeline (https://github.com/UniMelb-Day-Lab/DBLaCleaner) (*He et al., 2018*). This pipeline was used to de-multiplex and merge the paired-end reads, as well as remove low-quality sequences and chimeras using several filtering parameters (see *Appendix 1—figure 3* for additional details). These steps resulted in a total of 291,783 cleaned DBLα sequences for the 2572 *P. falciparum* isolates sequenced (*Appendix 1—figure 3*, *Appendix 1—table 3*). To identify the unique DBLα types, we then clustered these cleaned DBLα sequences with 241,693 DBLα sequences available from this Bongo study, as well as the *P. falciparum* reference strains (3D7, Dd2, and HB3) included as positive controls (*He et al., 2018*; *Pilosof et al., 2019*; *Rorick et al., 2018*), at the standard 96% sequence identity (https://github.com/UniMelb-Day-Lab/clusterDBLalpha) (*Barry et al., 2007*; *Day et al., 2017*, *Day et al., 2025*). Our dataset was then further curated by translating the DBLα types (N=68,503) into amino acid sequences and removing any DBLα types that could not be translated (i.e. contained a stop codon) (N=135; 0.2%). The remaining DBLα types were then assigned to their most likely DBLα domain class (i.e. DBLα, DBLα1, or DBLα2) using a hidden Markov model, and further classified based on the association of specific domain classes with semi-conserved upstream promoter sequences (ups) as either upsA or non-upsA DBLα types (https://github.com/UniMelb-Day-Lab/classifyDBLalpha)(*Ruybal-Pesántez et al., 2017*).

**Appendix 1—table 3.** The DBLα sequence pool sizes (upsA and non-upsA) and the number of unique DBLα types (upsA and non-upsA) observed in each survey (i.e. 2012, 2014, 2015, 2017) for those isolates with DBLα sequencing data.

| Survey | P. falciparum isolates sequenced | P. falciparum isolates with DBLα sequence data (≥1 DBLα type)* | P. falciparum isolates with DBLα sequence data (≥20 DBLα types)* | DBLα sequence pool size | | | Number of unique DBLα types | | |
|---|---|---|---|---|---|---|---|---|---|
| | | | | DBLα sequences | UpsA DBLα sequences[†] | Non-upsA DBLα sequences[‡] | DBLα types | UpsA DBLα types[§] | Non-upsA DBLα types[¶] |
| October 2012 (pre-IRS) | 808 | 742 (91.8) | 685 (84.8) | 120,029 | 22,881 (19.1) | 97,148 (80.9) | 35,377 | 2218 (6.3) | 33,159 (93.7) |
| October 2014 (IRS) | 430 | 386 (89.8) | 301 (70.0) | 33,489 | 7,048 (21.0) | 26,441 (79.0) | 16,334 | 1503 (9.2) | 14,831 (90.8) |
| October 2015 (post-IRS) | 545 | 510 (93.6) | 413 (75.8) | 42,774 | 8,942 (20.9) | 33,832 (79.1) | 19,584 | 1673 (8.5) | 17,911 (91.5) |
| October 2017 (SMC) | 789 | 759 (96.2) | 700 (88.7) | 92,757 | 18,625 (20.1) | 74,132 (79.9) | 29,423 | 2074 (7.0) | 27,349 (93.0) |
| TOTAL | 2,572 | 2,397 (93.2) | 2,099 (81.6) | 289,049 | 57,496 (19.9) | 231,553 (80.1) | 53,238 | 2802 (5.3) | 50,436 (94.7) |

Indoor residual spraying (IRS), seasonal malaria chemoprevention (SMC), interquartile range (IQR).

*Data reflect the number (% (n/N)) of P. falciparum isolates that had DBLα sequencing data relative to the number of participants sampled that were positive for P. falciparum by microscopy (including mixed P. falciparum infections). For those, a breakdown of those isolates by age group (years) with ≥20 DBLα types included in the analyses, see **Appendix 1—table 2**.

[†]Data reflect the upsA DBLα sequence pool size (% (n/N)) relative to the DBLα sequence pool size.

[‡]Data reflect the non-upsA DBLα sequence pool size (% (n/N)) relative to the DBLα sequence pool size.

[§]Data reflect the number (% (n/N)) of upsA DBLα types identified relative to the number of DBLα types identified.

[¶]Data reflect the number (% (n/N)) of non-upsA DBLα types identified relative to the number of DBLα types identified.

**Appendix 1—table 6.** The Kolmogorov-Smirnov test (KS-test) and the Pearson correlation tests (PC-test) are applied to compare the estimated MOI distributions for the population in each survey (i.e. 2012, 2014, 2015, 2017) based on either a uniform prior or a zero-truncated negative binomial prior with parameters in the ranges typical for low- (corresponding to a mean MOI~1.5), medium- (corresponding to a mean MOI~4.3), and high- (corresponding to a mean MOI~6.7) transmission endemic areas (see Materials and methods).

We present the results of comparison for both approaches which obtain the population-level MOI distribution from individual posterior MOI distributions, namely by pooling the maximum *a posteriori* MOI estimates (i.e. MAP pool) or by using mixture distribution (i.e. Mixture Dist.).

| Survey | Approach | Comparison (Negative binomial vs. Uniform) | KS-test statistics | KS-test p-value | PC-test statistics | PC-test p-value | Mean MOI difference* |
|---|---|---|---|---|---|---|---|
| October 2012 (pre-IRS) | MAP pool | Low vs. Uniform | 0.054015 | 0.036738 | 0.988473 | 0 | –0.32117 |
| | MAP pool | Medium vs. Uniform | 0.018978 | 0.966014 | 0.994902 | 0 | –0.08029 |
| | MAP pool | High vs. Uniform | 0.018978 | 0.966014 | 0.997470 | 0 | –0.00438 |
| October 2014 (IRS) | MAP pool | Low vs. Uniform | 0.036545 | 0.816283 | 0.990630 | 1.4E-260 | –0.13289 |
| | MAP pool | Medium vs. Uniform | 0.016611 | 0.999997 | 0.996893 | 0 | –0.00332 |
| | MAP pool | High vs. Uniform | 0.019934 | 0.999760 | 0.997409 | 0 | 0.01661 |
| October 2015 (post-IRS) | MAP pool | Low vs. Uniform | 0.046005 | 0.346351 | 0.984101 | 0 | –0.15496 |
| | MAP pool | Medium vs. Uniform | 0.009685 | 1 | 0.99500 | 0 | –0.02179 |
| | MAP pool | High vs. Uniform | 0.024213 | 0.968793 | 0.996081 | 0 | 0.01937 |
| October 2017 (SMC) | MAP pool | Low vs. Uniform | 0.055714 | 0.025924 | 0.985225 | 0 | –0.22714 |
| | MAP pool | Medium vs. Uniform | 0.011429 | 0.999989 | 0.994265 | 0 | –0.04714 |
| | MAP pool | High vs. Uniform | 0.011429 | 0.999989 | 0.996798 | 0 | –0.00286 |
| October 2012 (pre-IRS) | Mixture Dist. | Low vs. Uniform | 0.05387 | 0.037968 | – | – | –0.31321 |
| | Mixture Dist. | Medium vs. Uniform | 0.015199 | 0.997437 | – | – | –0.10165 |
| | Mixture Dist. | High vs. Uniform | 0.007393 | 1 | – | – | –0.02292 |

*Appendix 1—table 6 continued on next page*

Appendix 1—table 6 continued

| Survey | Approach | Comparison (Negative binomial vs. Uniform) | KS-test statistics | KS-test p-value | PC-test statistics | PC-test p-value | Mean MOI difference* |
|---|---|---|---|---|---|---|---|
| October 2014 (IRS) | Mixture Dist. | Low vs. Uniform | 0.040398 | 0.709761 | – | – | –0.18014 |
| | Mixture Dist. | Medium vs. Uniform | 0.011440 | 1 | – | – | –0.03705 |
| | Mixture Dist. | High vs. Uniform | 0.010149 | 1 | – | – | 0.00832 |
| October 2015 (post-IRS) | Mixture Dist. | Low vs. Uniform | 0.049573 | 0.263382 | – | – | –0.17625 |
| | Mixture Dist. | Medium vs. Uniform | 0.008205 | 1 | – | – | –0.02350 |
| | Mixture Dist. | High vs. Uniform | 0.010618 | 1 | – | – | 0.01732 |
| October 2017 (SMC) | Mixture Dist. | Low vs. Uniform | 0.048020 | 0.07961 | – | – | –0.24742 |
| | Mixture Dist. | Medium vs. Uniform | 0.009626 | 1 | – | – | –0.05289 |
| | Mixture Dist. | High vs. Uniform | 0.009381 | 1 | – | – | 0.00379 |

Indoor residual spraying (IRS), seasonal malaria chemoprevention (SMC), NA (–).

*The mean MOI difference is the difference in the mean value of the population-level MOI estimates from assuming either a zero-truncated negative binomial prior with different parameter choices or a uniform prior. Take the first row of the table as an example, the mean MOI difference value (–0.32117) is equal to the mean MOI for the population based on a zero-truncated negative binomial prior with a low-transmission parameter choice minus that based on a uniform prior. The order of comparison is consistent with the one listed in the Comparison column (Low vs. Uniform).

**Appendix 1—table 7.** The Kolmogorov-Smirnov test (KS-test) and the Pearson correlation tests (PC-test) are applied to compare the estimated MOI distributions for the population in each survey (i.e. 2012, 2014, 2015, 2017) between the zero-truncated negative binomial priors with parameters in the ranges typical for low- (corresponding to a mean MOI~1.5), medium- (corresponding to a mean MOI~4.3), and high- (corresponding to a mean MOI~6.7) transmission endemic areas (see Materials and methods).

We present the results of comparison for both approaches which obtain the population-level MOI distribution from individual posterior MOI distributions, namely by pooling the maximum *a posteriori* MOI estimates (i.e. MAP pool) or by using mixture distribution (i.e. Mixture Dist.).

| Survey | Approach | Comparison (Negative binomial) | KS-test statistics | KS-test p-value | PC-test statistics | PC-test p-value | Mean MOI difference* |
|---|---|---|---|---|---|---|---|
| October 2012 (pre-IRS) | MAP pool | Medium vs. Low | 0.051095 | 0.055938 | 0.986680 | 0 | 0.240876 |
| | MAP pool | Medium vs. High | 0.014599 | 0.998598 | 0.996125 | 0 | -0.075912 |
| | MAP pool | Low vs. High | 0.055474 | 0.029513 | 0.986516 | 0 | -0.316788 |
| October 2014 (IRS) | MAP pool | Medium vs. Low | 0.053156 | 0.362791 | 0.989269 | 8.011E-252 | 0.129568 |
| | MAP pool | Medium vs. High | 0.003322 | 1 | 0.998276 | 0 | -0.019934 |
| | MAP pool | Low vs. High | 0.056478 | 0.292237 | 0.988379 | 1.112E-246 | -0.149502 |
| October 2015 (post-IRS) | MAP pool | Medium vs. Low | 0.055690 | 0.154269 | 0.981405 | 9.764E-297 | 0.133172 |
| | MAP pool | Medium vs. High | 0.014528 | 0.999994 | 0.994613 | 0 | -0.041162 |
| | MAP pool | Low vs. High | 0.070218 | 0.034065 | 0.979963 | 3.920E-290 | -0.174334 |
| October 2017 (SMC) | MAP pool | Medium vs. Low | 0.044286 | 0.128371 | 0.983760 | 0 | 0.18 |
| | MAP pool | Medium vs. High | 0.011429 | 0.999989 | 0.995738 | 0 | -0.044286 |
| | MAP pool | Low vs. High | 0.055714 | 0.025924 | 0.982745 | 0 | -0.224286 |
| October 2012 (pre-IRS) | Mixture Dist. | Medium vs. Low | 0.040399 | 0.214223 | – | – | 0.211562 |
| | Mixture Dist. | Medium vs. High | 0.011531 | 0.999989 | – | – | -0.078734 |
| | Mixture Dist. | Low vs. High | 0.052049 | 0.049412 | – | – | -0.290295 |
| October 2014 (IRS) | Mixture Dist. | Medium vs. Low | 0.037816 | 0.785935 | – | – | 0.143090 |
| | Mixture Dist. | Medium vs. High | 0.011447 | 1 | – | – | -0.045364 |
| | Mixture Dist. | Low vs. High | 0.050535 | 0.425625 | – | – | -0.188454 |

*Appendix 1—table 7 continued on next page*

*Appendix 1—table 7 continued*

| Survey | Approach | Comparison (Negative binomial) | KS-test statistics | KS-test p-value | PC-test statistics | PC-test p-value | Mean MOI difference* |
|---|---|---|---|---|---|---|---|
| October 2015 (post-IRS) | Mixture Dist. | Medium vs. Low | 0.047779 | 0.303800 | – | – | 0.152752 |
| | Mixture Dist. | Medium vs. High | 0.010639 | 1 | – | – | –0.040816 |
| | Mixture Dist. | Low vs. High | 0.061610 | 0.087626 | – | – | –0.193568 |
| October 2017 (SMC) | Mixture Dist. | Medium vs. Low | 0.042425 | 0.161444 | – | – | 0.194537 |
| | Mixture Dist. | Medium vs. High | 0.0096285 | 0.999999 | – | – | –0.056674 |
| | Mixture Dist. | Low vs. High | 0.052140 | 0.044719 | – | – | –0.251211 |

Indoor residual spraying (IRS), seasonal malaria chemoprevention (SMC), NA (–).

*The mean MOI difference is the difference in the mean value of the population-level MOI estimates from assuming zero-truncated negative binomial priors with different parameter choices. Take the first row of the table as an example, the mean MOI difference value (0.240876) is equal to the mean MOI for the population based on a zero-truncated negative binomial prior with a medium-transmission parameter choice minus that based on a zero-truncated negative binomial prior with a low-transmission parameter choice. The order of comparison is consistent with the one listed in the Comparison column (Medium vs. Low).

**Appendix 1—table 1.** Age group (years) breakdown and parasitological characteristics of the participants surveyed in Bongo, Ghana, in each survey (i.e. 2012, 2014, 2015, 2017).

| | October 2012 (pre-IRS) | October 2014 (IRS) | October 2015 (post-IRS) | October 2017 (SMC) |
|---|---|---|---|---|
| Number of participants surveyed* | 1923 | 1866 | 2022 | 1915 |
| **Age groups (years)[†]** | | | | |
| Children: 1–5 years | 356 (18.5) | 216 (11.6) | 405 (20.0) | 354 (18.5) |
| Children: 6–10 years | 395 (20.5) | 421 (22.5) | 409 (20.2) | 358 (18.7) |
| Adolescents: 11–20 years | 413 (21.5) | 468 (25.1) | 467 (23.1) | 489 (25.5) |
| Adults: >20 years | 759 (39.5) | 761 (40.8) | 741 (36.7) | 714 (37.3) |
| **Microscopic *P. falciparum* prevalence[‡]** | 808 (42.0) | 430 (23.0) | 545 (27.0) | 789 (41.2) |
| Children: 1–5 years | 173 (48.6) | 37 (17.1) | 63 (15.6) | 49 (13.8) |
| Children: 6–10 years | 243 (61.5) | 142 (33.7) | 167 (40.8) | 184 (51.4) |
| Adolescents: 11–20 years | 202 (48.9) | 162 (34.6) | 169 (36.2) | 304 (62.2) |
| Adults: >20 years | 190 (25.0) | 89 (11.7) | 146 (19.7) | 295 (35.3) |
| **Microscopic *P. falciparum* density[§]** | 520 [160–1640] (40–126,040) | 200 [80–600] (40–73,360) | 320 [120-1800] (40–113,520) | 440 [160-1720] (40–97,520) |
| Children: 1–5 years | 1640 [400–9840] (40–126,040) | 440 [120–1080] (40–15,880) | 1840 [240–19,940] (40–113,520) | 4120 [760–20,800] (40–80,360) |
| Children: 6–10 years | 760 [240–1840] (40–61,560) | 320 [120–1280] (40–73,360) | 520 [200–2720] (40–48,600) | 1020 [320–5100] (40–97,520) |
| Adolescents: 11–20 years | 320 [160–760] (40–27,440) | 180 [80–240] (40–42,120) | 280 [120–1000] (40–42,880) | 440 [160–1210] (40–66,760) |
| Adults: >20 years | 200 [120–680] (40–31,040) | 120 [80–240] (40–41,320) | 120 [40–630] (40–40,280) | 220 [80–730] (40–19,040) |

Indoor residual spraying (IRS), seasonal malaria chemoprevention (SMC).

*Number of participants surveyed that were analysed by microscopy.

[†]Data reflect the number (% (n/N)) of participants surveyed in each age group.

[‡]Data reflect the number (% (n/N)) of participants surveyed that were microscopically positive for an asymptomatic *P. falciparum* infection (including mixed *P. falciparum* infections) relative to the number of participants surveyed in the total population and by the age groups presented.

[§]Median parasite density (i.e. parasites/µL of blood) (interquartile range [IQR]) (min-max) for the microscopically positive asymptomatic *P. falciparum* infections (including mixed *P. falciparum* infections).

**Appendix 1—table 2.** Microscopic *P. falciparum* DBLα type sequencing results, number of *P. falciparum var* repertoires (i.e. census population size), and mean MOI$_{var}$ by age group (years) in each survey (i.e. 2012, 2014, 2015, 2017).

| | October 2012 (pre-IRS) | October 2014 (IRS) | October 2015 (post-IRS) | October 2017 (SMC) |
|---|---|---|---|---|
| Number of microscopic *P. falciparum* isolates | 808 | 430 | 545 | 789 |
| *P. falciparum* isolates with DBLα sequencing data (≥20 DBLα types)* [§] | 685 (84.8) | 301 (70.0) | 413 (75.8) | 700 (88.7) |

*Appendix 1—table 2 Continued on next page*

*Appendix 1—table 2 Continued*

| | October 2012 (pre-IRS) | October 2014 (IRS) | October 2015 (post-IRS) | October 2017 (SMC) |
|---|---|---|---|---|
| Children: 1–5 years | 158 (91.3) | 28 (75.7) | 51 (81.0) | 44 (89.8) |
| Children: 6–10 years | 217 (89.3) | 116 (81.7) | 146 (87.4) | 170 (92.4) |
| Adolescents: 11–20 years | 167 (82.7) | 112 (69.1) | 129 (76.3) | 284 (93.4) |
| Adults: >20 years | 143 (75.3) | 45 (50.6) | 87 (59.6) | 202 (68.5) |
| Number of *P. falciparum var* repertoires (i.e. census population size)[†][§] | 2552 (2354–2756) | 731 (628 - 836) | 909 (804–1019) | 2087 (1926–2254) |
| Children: 1–5 years | 495 (402–595) | 80 (43–125) | 62 (39–88) | 90 (54–132) |
| Children: 6–10 years | 1035 (908–1166) | 318 (247–397) | 378 (311–451) | 750 (642–861) |
| Adolescents: 11–20 years | 683 (579–792) | 256 (200–315) | 313 (250–382) | 827 (734–921) |
| Adults: >20 years | 338 (278–401) | 76 (48–111) | 155 (119–195) | 420 (365–478) |
| *P. falciparum* mean $MOI_{var}$ [‡][§] | 4.38 (4.16–4.61) | 2.74 (2.49–3.02) | 2.57 (2.40–2.77) | 3.28 (3.12–4.45) |
| Children: 1–5 years | 5.16 (4.68–5.67) | 2.86 (1.93–4.04) | 2.27 (1.98–2.59) | 3.34 (2.73–4.05) |
| Children: 6–10 years | 5.26 (4.87–5.67) | 3.22 (2.76–3.72) | 2.96 (2.64–3.32) | 4.41 (4.00–4.84) |
| Adolescents: 11–20 years | 4.09 (3.68–4.52) | 2.59 (2.25–2.96) | 2.74 (2.40–3.12) | 3.45 (3.21–3.70) |
| Adults: >20 years | 2.52 (2.27–2.78) | 1.80 (1.38–2.38) | 1.85 (1.62–2.11) | 2.08 (1.93–2.24) |

*Data reflect the number (% (n/N)) of *P. falciparum* isolates that had DBLα sequencing data relative to the number of participants sampled that were positive for *P. falciparum* by microscopy (including mixed *P. falciparum* infections) in the total population and by the age groups (see **Appendix 1—table 1** for total number of participants sampled that were microscopy positive).

[†]Number of *var* repertoires (i.e. census population size) (95% confidence interval). To account for differences in sampling depth across age groups and surveys, we performed subsampling with replacement by selecting the minimum number of individuals in each age group across all surveys. We then calculated the total number of *var* repertoires from these subsampled individuals within each age group in each survey. This approach ensures consistent sample sizes within each age group across all surveys. Finally, we summed the *var* repertoires across age groups to obtain the total *var* repertoire count for each survey. The mean and 95% CIs for the number of *var* repertoires (i.e. census population size) were estimated by repeating the subsampling procedure 10,000 times. The CIs were then derived from the distribution of these repeated subsampling replicates.

[‡]Mean $MOI_{var}$ (95% confidence interval) based on pooling the maximum *a posteriori* MOI estimates. The 95% confidence intervals (95% CIs) were calculated based on a bootstrap approach. We resampled 10,000 replicates from the original population-level MOI distribution with replacement. Each resampled replicate has the same size as the original sample. We then derive the 95% CI based on the distribution of the resampled replicates.

[§]*Note*: There is a simple map between census population size and mean MOI, as one can simply divide or multiply by the sample size (i.e. the number of *P. falciparum* isolates with DBLα sequencing data), respectively, to convert between the two quantities.

**Appendix 1—table 4.** The Kolmogorov-Smirnov test (KS-test) is applied to compare the estimated multiplicity of infection (MOI) distributions for the population in each survey (i.e. 2012, 2014, 2015,

2017) for the two approaches, namely by pooling the maximum *a posteriori* MOI estimates or by using the mixture distributions (see Materials and methods) for the uniform prior.

| Survey | Prior | KS-test statistics | p-value | Mean MOI difference* |
|---|---|---|---|---|
| October 2012 (pre-IRS) | Uniform | 0.019592 | 0.955175 | –0.13235 |
| October 2014 (IRS) | Uniform | 0.030432 | 0.943197 | –0.13112 |
| October 2015 (post-IRS) | Uniform | 0.038248 | 0.581476 | –0.1149 |
| October 2017 (SMC) | Uniform | 0.023823 | 0.821812 | –0.1298 |

Indoor residual spraying (IRS), seasonal malaria chemoprevention (SMC).

*The mean MOI difference is the difference in the mean value of the population-level MOI estimates from either pooling the maximum *a posteriori* estimates or the mixture distribution.

**Appendix 1—table 5.** The Kolmogorov-Smirnov test (KS-test) is applied to compare the estimated multiplicity of infection (MOI) distributions for the population in each survey (i.e. 2012, 2014, 2015, 2017) for the two approaches, namely by pooling the maximum *a posteriori* MOI estimates or by using the mixture distributions (see Materials and methods) using the negative binomial distribution with parameters in the range typical for low- (corresponding to a mean MOI~1.5), medium- (corresponding to a mean MOI~4.3), and high- (corresponding to a mean MOI~6.7) transmission endemic areas.

| Survey | Negative binomial | KS-test statistics | p-value | Mean MOI difference* |
|---|---|---|---|---|
| October 2012 (pre-IRS) | Low | 0.024189 | 0.817681 | –0.14031 |
| | Medium | 0.021339 | 0.914045 | –0.11099 |
| | High | 0.019512 | 0.956709 | –0.11381 |
| October 2014 (IRS) | Low | 0.027862 | 0.973588 | –0.08387 |
| | Medium | 0.032505 | 0.908128 | –0.09739 |
| | High | 0.036612 | 0.814540 | –0.12282 |
| October 2015 (post-IRS) | Low | 0.037872 | 0.594217 | –0.09361 |
| | Medium | 0.032974 | 0.760228 | –0.11319 |
| | High | 0.037138 | 0.619209 | –0.11284 |
| October 2017 (SMC) | Low | 0.025665 | 0.745796 | –0.10952 |
| | Medium | 0.026964 | 0.68884 | –0.12406 |
| | High | 0.030275 | 0.542523 | –0.13645 |

Indoor residual spraying (IRS), seasonal malaria chemoprevention (SMC).

*The mean MOI difference is the difference in the mean value of the population-level MOI estimates from either pooling the maximum *a posteriori* estimates or the mixture distribution for different parameter choices of a zero-truncated negative binomial prior.

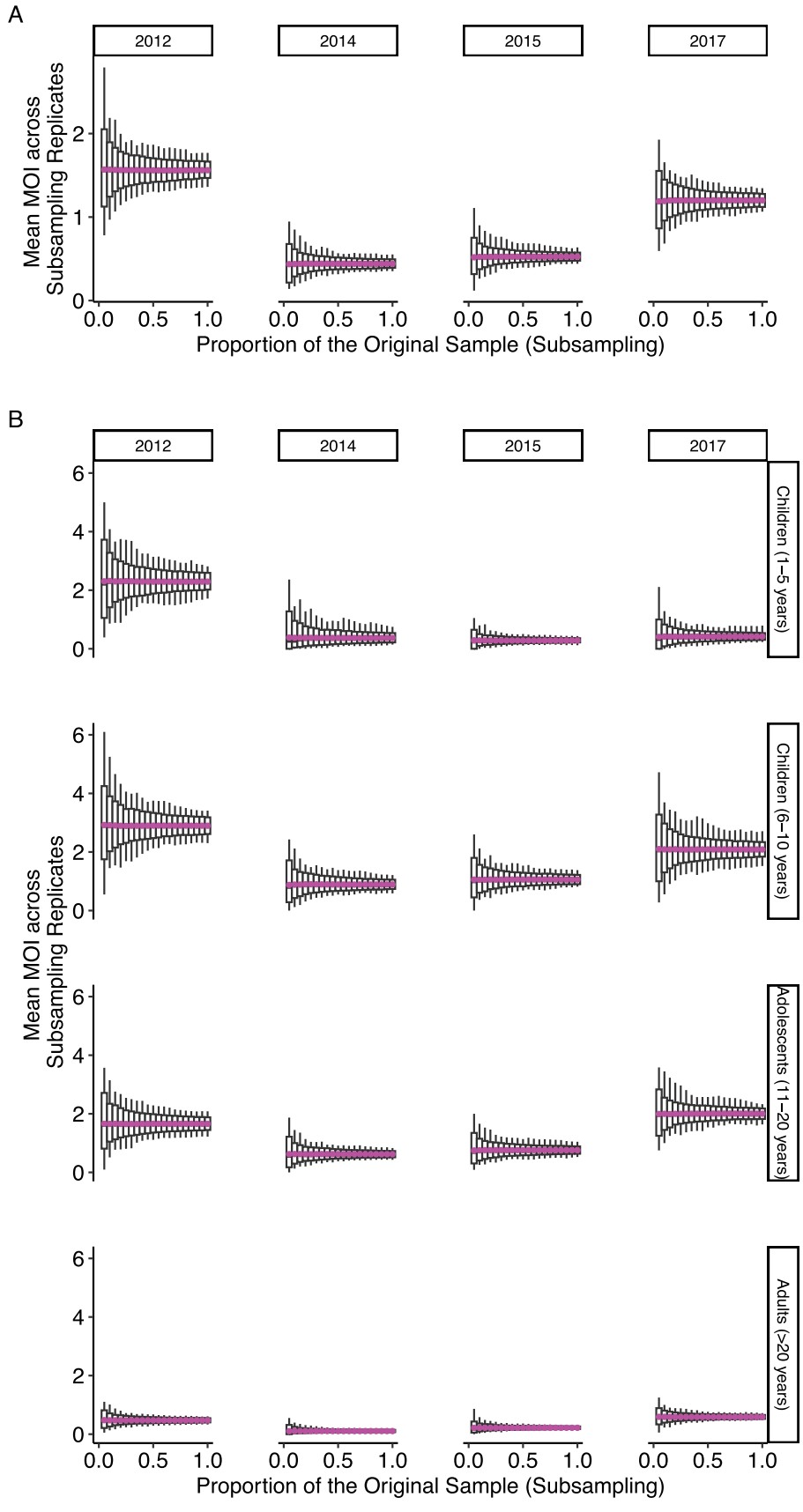

**Appendix 1—figure 1.** Mean multiplicity of infection (MOI) averaged over all sampled individuals for the subsampling replicates in 2012 (pre-indoor residual spraying [IRS]), 2014 (during IRS), 2015 (post-IRS), and 2017 (seasonal malaria chemoprevention [SMC]). The mean MOI (pink dots) for the (**A**) study population and (**B**) for all age groups (years) in each survey is stable relative to the sampling size or sampling depth, which allows for the extrapolation from the census population size of our population sample to that of the whole population of local hosts. For each sampling depth, we generate 1000 subsampling replicates with replacement. Minimum, 5% quantile, median, 95% quantile, and maximum values are shown in the boxplot.

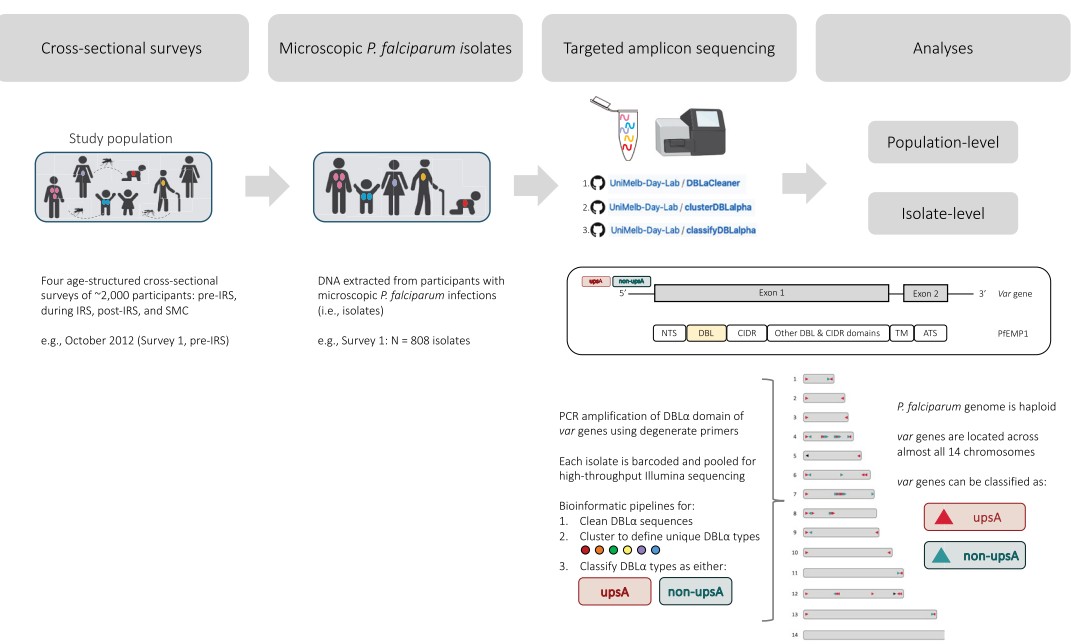

**Appendix 1—figure 2.** Schematic diagram of the *var* genotyping (i.e. *var*coding) approach. For additional details about each step, see Materials and methods and Appendix 1. A schematic insert of the *var* gene locus and PfEMP1 has been included as defined in *Rask et al., 2010*, with its N-terminal segment (NTS), Duffy binding-like (DBL) domains, cysteine-rich interdomain regions (CIDR), one transmembrane region (TM), and the acidic terminal segment (ATS). The Illumina MiSeq sequencer stock image was created using BioRender.com.

93,874,628 paired reads

Minimum read length 100nt
Maximum uncalled bases 15
Merge paired-end reads
Minimum assembly length 100nt
Minimum overlap between a read pair 10nt

86,610,028 merged sequences

Filter low-quality sequences with > one expected error
Filter chimeras

58,212,163 sequences

Remove singletons
Cluster reads at 96% sequence identity

2,092,592 clusters

Remove low-support clusters with <15 sequences
Keep representative sequence from each cluster
Filter non-DBLα sequences using domain threshold score of 80
Align to 3D7, Dd2 and HB3 reference as quality check

291,783 cleaned DBLα sequences

**Appendix 1—figure 3.** Bioinformatic sequence data processing flowchart. The flowchart shows the bioinformatic process to clean the raw de-multiplexed paired reads, with details on the filtering parameters utilised at each step. This customised bioinformatic pipeline is described in detail in *He et al., 2018*.

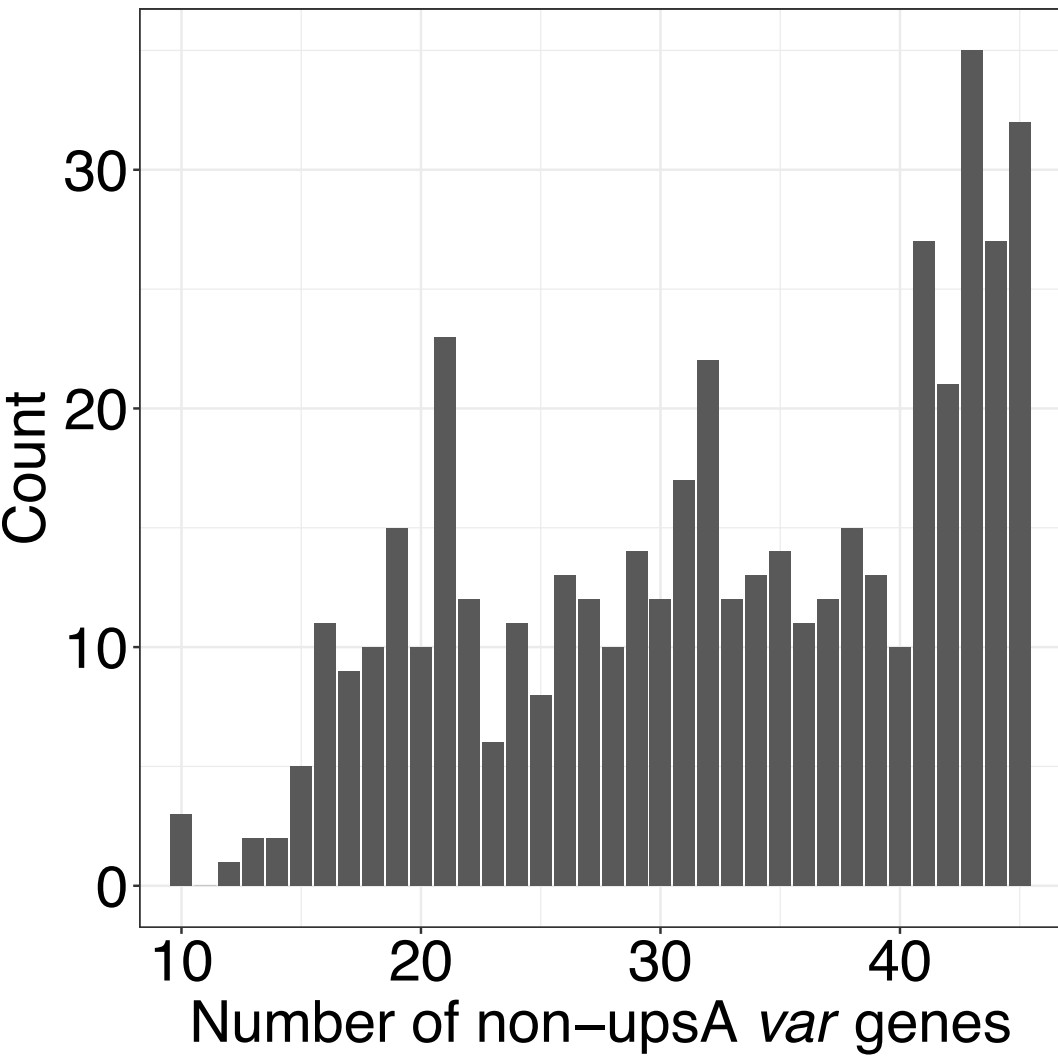

**Appendix 1—figure 4.** Histogram of the number of non-upsA DBLα *var* gene types sequenced per repertoire for those isolates with monoclonal infections (multiplicity of infection [MOI] =1) (*Labbé et al., 2023*). The molecular sequences used to derive this repertoire size distribution were previously sequenced from isolates sampled during six cross-sectional surveys made from 2012 to 2016 in Bongo District, Ghana (*He et al., 2018*; *Pilosof et al., 2019*; *Ruybal-Pesántez et al., 2022*; *Tiedje et al., 2022*). These isolates were estimated to be monoclonal infections (i.e. human hosts estimated to be infected by a single *P. falciparum* clone, MOI=1), based on a cut-off value of 45 non-upsA DBLα types. This cut-off was selected based on the median number of non-upsA DBLα types identified for the 3D7 laboratory isolate included as a control during *varcoding* (*Ghansah et al., 2023*). A version of this figure was previously published (*Labbé et al., 2023*, Figure 2; CC BY 4.0 licence). The copyright holder has granted permission to publish under a CC BY 4.0 licence.

