## [Editor Report · eLife Assessment]

This **valuable** study highlights how the diversity of the malaria parasite population diminishes following the initiation of effective control interventions but quickly rebounds as control wanes. It also demonstrates that the asymptomatic reservoir is unevenly distributed across host age groups. The data presented are **convincing** and the work shows how genetic studies could be used to monitor changes in disease transmission.

---

## [Referee Report · Reviewer #2 (Public review)]

In this manuscript, Tiedje and colleagues longitudinally track changes in parasite number across four time points as a way of assessing the effect of malaria control interventions in Ghana. Some of the study results have been reported previously, and in this publication, the authors focus on age-stratification of the results. Malaria prevalence was lower in all age groups after IRS. Follow-up with SMC, however, maintained lower parasite prevalence in the targeted age group but not the population as a whole. Additionally, they observe that diversity measures rebound more slowly than prevalence measures. This adds to a growing literature that demonstrates the relevance of asymptomatic reservoirs.

Overall, I found these results clear, convincing, and well presented. There is growing interest in developing an expanded toolkit for genomic epidemiology in malaria, and detecting changes in transmission intensity is one major application. As the authors summarize, there is no one-size-fits-all approach, and the Bayesian MOIvar estimate developed here has the potential to complement currently used methods, particularly in regions with high diversity/transmission. I find its extension to a calculation of absolute parasite numbers appealing as this could serve as both a conceptually straightforward and biologically meaningful metric.

As the authors address, their use of the term "census population size" is distinct from how the term is used in the population genetics literature. I therefore anticipate that parasite count will be most useful in an epidemiological context where the total number of sampled parasites can be contrasted with other metrics to help us better understand how parasites are divided across hosts, space, and time.

---

## [Author Response]

The following is the authors’ response to the previous reviews

**Reviewer #2 (Public review):**
In this manuscript, Tiedje and colleagues longitudinally track changes in parasite numbers across four time points as a way of assessing the effect of malaria control interventions in Ghana. Some of the study results have been reported previously, and in this publication, the authors focus on age-stratification of the results. Malaria prevalence was lower in all age groups after IRS. Follow-up with SMC, however, maintained lower parasite prevalence in the targeted age group but not the population as a whole. Additionally, they observe that diversity measures rebound more slowly than prevalence measures. This adds to a growing literature that demonstrates the relevance of asymptomatic reservoirs.Strengths:Overall, I found these results clear, convincing, and well-presented. There is growing interest in developing an expanded toolkit for genomic epidemiology in malaria, and detecting changes in transmission intensity is one major application. As the authors summarize, there is no one-size-fits-all approach, and the Bayesian MOIvar estimate developed here has the potential to complement currently used methods, particularly in regions with high diversity/transmission. I find its extension to a calculation of absolute parasite numbers appealing as this could serve as both a conceptually straightforward and biologically meaningful metric.

We thank the reviewer for this positive review of our results and approach.

Weaknesses:While I understand the conceptual importance of distinguishing among parasite prevalence, mean MOI, and absolute parasite number, I am not fully convinced by this manuscript's implementation of "census population size".

This reviewer remains unconvinced of the use of the term “census population size”. This appears to be due to the dependence of the term on sample size rather than representing a count of a whole population. To give context to our use we are clear in the study presented that the term describes a count of the parasite “strains” in an age-specific sample of a human population in a specified location undergoing malaria interventions.

They have suggested instead using “sample parasite count”. We argue that this definition is too specific and less applicable when we extrapolate the same concept to a different denominator, such as the population in a given area. Importantly, our ecological use of a census allows us to count the appearance of the same strain more than once should this occur in different people.

The authors reference the population genetic literature, but within the context of that field, "census population size" refers to the total population size (which, if not formally counted, can be extrapolated) as opposed to "effective population" size, which accounts for a multitude of demographic factors. There is often interesting biology to be gleaned from the magnitude of difference between N and Ne.

As stated in the introduction we have been explicit in saying that we are not using a population genetic framework. Exploration of N and Ne in population genetics has merit. How this is reconciled when using a “strain” definition and not neutral markers would need to be assessed.

In this manuscript, however, "census population size" is used to describe the number of distinct parasites detected within a sample, not a population. As a result, the counts do not have an immediate population genetic interpretation and cannot be directly compared to Ne. This doesn't negate their usefulness but does complicate the use of a standard population genetic term.

We are clear we are defining a census of parasite strains in an age-specific sample of a population living in two catchment areas of Bongo District. We appreciate the concern of the reviewer and have now further edited the relevant paragraphs in both the Introduction (Lines 75-80) and the Discussion (Lines 501-506) to make very clear the dependence of the reported quantity on sample size, but also its feasible extrapolation consistent with the census of a population.

In contrast, I think that sample parasite count will be most useful in an epidemiological context, where the total number of sampled parasites can be contrasted with other metrics to help us better understand how parasites are divided across hosts, space and time. However, for this use, I find it problematic that the metric does not appear to correct for variations in participant number. For instance, in this study, participant numbers especially varied across time for 1-5 year-olds (N=356, 216, 405, and 354 in 2012, 2014, 2015, and 2017 respectively).

The reviewer has made an important point that for the purpose of comparisons across the four surveys or study time points (i.e. 2012, 2014, 2015, and 2017), we should "normalize" the number of individuals considered for the calculation of the "census population size". Given that this quantity is a sum of the estimated MOI_var,,_ we need to have constant numbers for its values to be compared across the surveys, within age group and the whole population. This is needed not only to get around the issue of the drop in 1-5 year olds surveyed in 2014 but to also stabilize the total number of individuals for the whole sample and for specific age groups. One way to do this is to use the smaller sample size for each age group across time, and to use that value to resample repeatedly for that number of individuals for surveys where we have a larger sample size. This has now been updated included in the manuscript as described in the Materials and Methods (Lines 329-341) and in the Results (Lines 415-430; see updated Figure 4 and Table supplement 7). This correction produces very similar results to those we had presented before (see updated Figure 4 and Table supplement 7).

As stated in our previous response we have used participant number in an interrupted time series where the population was sampled by age to look at age-specific effects of sequential interventions IRS and SMC. As shown in Table supplement 1 of the 16 age-specific samples of the total population, we have sampled very similar proportions of the population by age group across the four surveys. The only exception was the 1-5 year-old age group during the survey in 2014. We are happy to provide additional details to further clarify the lower number (or percentage) of 1-5 year olds (based on the total number of participants per survey) in 2014 (~12%; N = 216) compared to the other surveys conducted 2012, 2015, and 2017 (~18-20%; N = 356, 405, and 354, respectively). Please see Table supplement 1 for the total number of participants surveyed in each of the four surveys (i.e. 2012, 2014, 2015, and 2017).

This sample size variability is accounted for with other metrics like mean MOI.

We agree that mean MOI by age presents a way forward with variable samples to scale up. Please see updated Figure supplement 8.

In sum, while the manuscript opens up an interesting discussion, I'm left with an incomplete understanding of the robustness and interpretability of the new proposed metric.”

We thank you for your opinion. We have further edited the manuscript to make clear our choice of the term and the issue of sample size. We believe the proposed terminology is meaningful as explained above.

**Reviewer #3 (Public review):**
SummaryThe manuscript coins a term "the census population size" which they define from the diversity of malaria parasites observed in the human community. They use it to explore changes in parasite diversity in more than 2000 people in Ghana following different control interventions.Strengths:This is a good demonstration of how genetic information can be used to augment routinely recorded epidemiological and entomological data to understand the dynamics of malaria and how it is controlled. The genetic information does add to our understanding, though by how much is currently unclear (in this setting it says the same thing as age stratified parasite prevalence), and its relevance moving forward will depend on the practicalities and cost of the data collection and analysis. Nevertheless, this is a great dataset with good analysis and a good attempt to understand more about what is going on in the parasite population.

Thank you to the reviewer for their supportive assessment of our research.

WeaknessesNoneReview**er #3 (Recommendations for the authors):**New figure supplement 8 - x-axis says percentage but goes between 0-1, so is a proportion

We thank the reviewer for bringing this to our attention. We have amended the x-axis labels accordingly for Figure supplement 8.